# Antibiotic resistance plasmids in *Enterobacteriaceae* isolated from fresh produce in northern Germany

Maria Stein,[1] Erik Brinks,[1] Jannike Loop,[1] Diana Habermann,[1] Gyu-Sung Cho,[1] Charles M. A. P. Franz[1]

**ABSTRACT**   In this study, the genomes of 22 *Enterobacteriaceae* isolates from fresh produce and herbs obtained from retail markets in northern Germany were completely sequenced with MiSeq short-read and MinION long-read sequencing and assembled using a Unicycler hybrid assembly. The data showed that 17 of the strains harbored between one and five plasmids, whereas in five strains, only the circular chromosomal DNA was detected. In total, 38 plasmids were identified. The size of the plasmids detected varied between ca. 2,000 and 326,000 bp, and heavy metal resistance genes were found on seven (18.4%) of the plasmids. Eleven plasmids (28.9%) showed the presence of antibiotic resistance genes. Among large plasmids (>32,000 bp), IncF plasmids (specifically, IncFIB and IncFII) were the most abundant replicon types, while all small plasmids were Col-replicons. Six plasmids harbored unit and composite transposons carrying antibiotic resistance genes, with IS26 identified as the primary insertion sequence. Class 1 integrons carrying antibiotic resistance genes were also detected on chromosomes of two *Citrobacter* isolates and on four plasmids. Mob-suite analysis revealed that 36.8% of plasmids in this study were found to be conjugative, while 28.9% were identified as mobilizable. Overall, our study showed that *Enterobacteriaceae* from fresh produce possess antibiotic resistance genes on both chromosome and plasmid, some of which are considered to be transferable. This indicates the potential for *Enterobacteriaceae* from fresh produce that is usually eaten in the raw state to contribute to the transfer of resistance genes to bacteria of the human gastrointestinal system.

**IMPORTANCE** This study showed that *Enterobacteriaceae* from raw vegetables carried plasmids ranging in size from 2,715 to 326,286 bp, of which about less than one-third carried antibiotic resistance genes encoding resistance toward antibiotics such as tetracyclines, aminoglycosides, fosfomycins, sulfonamides, quinolones, and β-lactam antibiotics. Some strains encoded multiple resistances, and some encoded extended-spectrum β-lactamases. The study highlights the potential of produce, which may be eaten raw, as a potential vehicle for the transfer of antibiotic-resistant bacteria.

**KEYWORDS**   fresh produce, antibiotic resistance, whole genome sequencing, plasmids, *Enterobacteriaceae*

Antibiotic resistance is a global health and developmental threat. As antibiotics are becoming increasingly ineffective, resistant bacteria can spread globally and lead to infections that are difficult to treat and may result in death (1, 2). One possible source of antibiotic-resistant pathogens and opportunistic pathogens is food, in particular, animal meats (3). In contrast to food products originating from animals, fresh produce has also come into focus as a reservoir for antibiotic-resistant and/or pathogenic bacteria (4–7). Vegetables, fruits, fresh-cut products, and sprouts are considered part of a healthy diet because they supply a combination of vitamins, antioxidants, and minerals (8). However, because of the heat instability of the nutritional compounds, produce is often

Address correspondence to Gyu-Sung Cho, gyusung.cho@mri.bund.de.

The authors declare no conflict of interest.

consumed minimally processed or raw, and contaminating bacteria are, therefore, not inactivated (9). Fresh produce can, therefore, cause widespread disease outbreaks when it is contaminated with pathogens along the farm-to-fork route (8). The most common human pathogenic bacteria associated with fresh produce are *Escherichia* (*E.*) *coli, Listeria* (*L.*) *monocytogenes,* and *Salmonella* (8, 10). For example, *Salmonella* Coeln caused an outbreak associated with ready-to-eat salads in Norway (11), fresh-bagged spinach contaminated by *E. coli* O157:H7 led to a multistate outbreak in the USA in 2006 (12), and packaged leafy greens contaminated with *L. monocytogenes* caused a listerioses outbreak in USA and Canada (13). In Germany, unprocessed fresh produce was brought more into public focus as a potential vehicle for pathogenic bacteria after the *E. coli* O104:H4 outbreak caused by contaminated sprouts in northern Germany in 2011, where 54 people died (14, 15).

Fiedler et al. (16) examined 200 fresh produce samples from markets in northern Germany and reported a low incidence of pathogenic bacteria in these products. Despite this, the mean aerobic mesophilic bacterial counts were considerably high (7 to 8 $\log_{10}$ cfu/g), and the *Enterobacteriaceae* counts varied greatly within the sampled products and ranged from 2 to 7.5 $\log_{10}$ cfu/g in leaf lettuce (16). This indicates that fresh produce may be a potential vehicle for opportunistic pathogens, including potentially antibiotic-resistant *Enterobacteriaceae* that may contribute to the spread of antibiotic resistance genes via this food route (17, 18). In a previous study, Blau et al. (19) isolated tetracycline-resistant *E. coli* from mixed salads, cilantro, and arugula from the German market and showed that these carried IncF, IncI1, IncN, IncHI1, IncU, IncP-1 β, and IncX1 plasmids. Furthermore, conjugative plasmids encoding tetracycline resistance were captured by exogenous plasmid isolation using an *E. coli* recipient strain (19). The study, thus, emphasized the role of conjugative plasmids in *Enterobacteriaceae* in horizontal gene transfer that might take place on fresh produce, adding to the spread of antibiotic resistance genes.

Our previous study showed the presence of potentially opportunistic pathogens, belonging mostly to *E. coli, Klebsiella* (*K.*) *pneumoniae, Citrobacter* (*C.*) *portucalensis,* and *Enterobacter* (*En.*) *ludwigii,* isolated from fresh produce in Germany. Among these, strains resistant to multiple antibiotic classes were identified (18). In this study, we aimed to obtain complete chromosome and plasmid sequences from 22 strains isolated from fresh produce using a hybrid assembly approach, which combined short-read (MiSeq) and long-read (MinION) sequencing data. Additionally, we utilize various databases including ResFinder, PlasmidFinder, and Mob-suite to characterize mobile genetic elements (MGEs), particularly plasmids, within the complete genome sequences. This investigation, thus, aimed to gain a better understanding of the diversity of antibiotic resistance genes and their genetic location in strains of fresh produce origin. Furthermore, we aimed to investigate the diversity of the plasmids present in these strains and the potential role of the plasmids in spreading antibiotic resistance genes across the farm-to-fork route.

## RESULTS

### Complete genome sequences and precise identification of all strains

In this study, previously sequenced strains using Illumina Miseq, along with additional strains isolated from fresh produces, were sequenced using the Oxford Nanopore Technologies MinION long-read sequencing platform. The combined data from long- and short-read sequencing were assembled by a hybrid assembly method. Using this approach, we were able to fully resolve the chromosomes of 22 *Enterobacteriaceae* strains, as well as all of the 38 plasmids and one extrachromosomal prophage harbored by the strains.

The results in Table 1 confirm the findings of the previous study, where MiSeq data were used to identify strains using various genotyping and whole genome analysis approaches, excluding strains Cif11, Kva3, Kpneu8, Kpneu28, and Kpneu34, which were newly determined in this study (18). Additional information obtained from complete

**TABLE 1** Species identification and source of isolation of all included strains[b]

| Strain no. | Genome comparison with type or reference strain (% dDDH) | Source | Country of product origin |
|---|---|---|---|
| Cigi1 | *Citrobacter gillenii* AF64_5pH9A[a] (92.5) | Arugula | Germany |
| Cipo4 | *Citrobacter portucalensis* A60[T] (80.2) | Mixed salad (carrots, leeks, celery) | Unknown origin |
| Ciw5.1 | *Citrobacter werkmanii* NBRC 105721[T] (70.1) | China rose sprouts, radish sprouts | Netherlands |
| Ciw5.2 | *Citrobacter werkmanii* NBRC 105721[T] (70.1) | China rose sprouts, radish sprouts | Netherlands |
| Cipa6.1 | *Citrobacter pasteurii* CIP 55.13[T] (94.3) | Mung bean sprouts | Germany |
| Cipa6.2 | *Citrobacter pasteurii* CIP 55.13[T] (94.3) | Mung bean sprouts | Germany |
| Cif11 | *Citrobacter freundii* ATCC 8090[T] (91.7) | China rose sprouts, radish sprouts | Netherlands |
| Cipo13 | *Citrobacter portucalensis* A60[T] (78.7) | Arugula | Italy |
| Endy1 | *Enterobacter dykesii* DSM 111347[T] (100) | Mung bean sprouts | Germany |
| Endy2 | *Enterobacter dykesii* DSM 111347[T] (100) | Mung bean sprouts | Germany |
| Enh11 | *Enterobacter hormaechei* subsp. *steigerwaltii* DSM16691[T] (91.8) | Cucumber | Germany |
| Enb12 | *Enterobacter bugandensis* EB247[T] (84.3) | Chickpea sprouts, azuki sprouts, bean sprouts | Netherlands |
| Ec1115 | *Escherichia coli* DSM30083[T] (75.1) | Lollo rosso, lollo biondo | Germany |
| Ec1117 | *Escherichia coli* DSM30083[T] (74.7) | Oregano | Netherlands |
| Ec1119 | *Escherichia coli* DSM30083[T] (74.1) | Carrots | Germany |
| Ec1120 | *Escherichia coli* DSM30083[T] (75.7) | Lollo rosso, lollo biondo | Germany |
| Kgr1 | *Klebsiella grimontii* 06D021[T] (95.5) | Carrots | Germany |
| Kva3 | *Klebsiella variicola* DSM 15968[T] (92.7) | Marjoram | Kenya |
| Kpneu4 | *Klebsiella pneumoniae* ATCC 13883[T] (92.9) | Carrots | Germany |
| Kpneu8 | *Klebsiella pneumoniae* ATCC 13883[T] (93.8) | Onion sprouts | Netherlands |
| Kpneu28 | *Klebsiella pneumoniae* ATCC 13883[T] (93.1) | Organic sprouts of alfalfa, radishes, broccoli | Germany |
| Kpneu34 | *Klebsiella pneumoniae* ATCC 13883[T] (92.9) | Organic sprouts of mini mung beans, chickpeas, adzuki beans, green peas, wheat | Germany |

[a]Reference strain, draft genome of the *C. gillenii* type strain is not available.
[b]Cif, *Citrobacter freundii*; Cigi, *Citrobacter gillenii*; Cipo, *Citrobacter portucalensis*; Cipa, *Citrobacter pasteurii*; Ciw, *Citrobacter werkmanii*; Enb, *Enterobacter bugandensis*; Endy, *Enterobacter dykesii*; Enh, *Enterobacter hormaechei*; Ec, *Escherichia coli*; Kgr, *Klebsiella grimontii*; Kpneu, *Klebsiella pneumoniae; Kva, Klebsiella variicola*.

genome sequence data, such as total coding sequences (CDSs), number of contigs, contig length, and mol% GC contents, is presented in Table 2. For precise bacterial identification using *in silico* digital DNA:DNA hybridization (dDDH), draft genome sequence data were deemed to be sufficient, as was observed also in a previous study (20). The sequencing by both long- and short-read methods suggested that strains *Enterobacter* (*En*.) *dykesii* Endy1 and *En. dykesii* Endy2 were very similar, and dDDH values showed 100% identity. In addition, both strains were also isolated from the same fresh produce (Table 1), and therefore, these appeared to probably represent clonal isolates. Strain *Citrobacter* (*C*.) *gillenii* Cigi1 was compared by dDDH to the *C. gillenii* AF64_5pH9A strain, for which a genome sequence was available (October 2023). This comparison showed a 92.5% sequence similarity (Table 1), indicating that strain Cigi1 could also be identified as *C. gillenii*. The 22 *Enterobacteriaceae* strains of this study, thus, consisted of *Citrobacter* spp. (*n* = 8), *Enterobacter* spp. (*n* = 4), *Escherichia coli* (*n* = 4), and *Klebsiella* spp. (*n* = 6) (Table 1). All of these genera are known to contain species of importance as opportunistic nosocomial pathogens and are often antibiotic-resistant (21).

## Phenotypic antibiotic resistance

The phenotypic antibiotic resistances are shown in Table 3. Nineteen out of 22 strains (86%) showed resistance to ampicillin, while two strains (9%) showed an intermediate resistance. Thus, only one strain, i.e., *E. coli* Ec1120, was susceptible to ampicillin. The second most commonly occurring resistance was against tetracycline, as 12 of the 22 strains (55%) were resistant. Eight strains (36%) showed resistance to streptomycin, and seven strains (32%) to ciprofloxacin (Table 3). Four strains (18%) showed resistance to chloramphenicol and cefotaxime, and two strains to gentamicin (9%). Eleven of

**TABLE 2** Results of bioinformatic analyses (e.g., ResFinder, PlasmidFinder, and Mob-suite) of complete chromosome and plasmid sequences of all strains relating to genome characteristics, antibiotic resistance genes, heavy metal resistance genes, and predicted mobility

| Strain name/ chrom. | Plasmid | Total CDSs | PlasmidFinder (Mob-suite rep-type) | Sequence length (bp) | mol% GC content | Resistance genes (metal resistance genes) | Phenotypic resistance | Predicted mobility (mpf type) |
|---|---|---|---|---|---|---|---|---|
| Cigi1 | | 5,018 | | 5,035,347 | 52.50 | | TC, STR (CL), AMP | |
| | pCIGI1_1 | | | 119,408 | 54.22 | *tet*(Y), *aph* (6)-Id, *aph*(3″)-Ib | | Conjugative (MPF$_F$) |
| | pCIGI1_2 | | IncFIB (IncFIB) | 107,458 | 50.87 | | | Non-mobilizable |
| | pCIGI1_3 | | | 55,797 | 51.89 | | | Mobilizable |
| Cipo4 | | 4,706 | | 4,909,244 | 51.88 | *qnr*B9, *bla*$_{CMY-2}$ | TC, STR, AMP | |
| | pCIPO4_1 | | | 91,909 | 52.06 | *tet*(D) | | Mobilizable |
| | pCIPO4_2 | | IncR, (IncR) | 49,031 | 51.74 | *tet*(A) | | Mobilizable |
| Ciw5.1 | | 4,824 | | 5,085,698 | 52.09 | *aad*A1, *bla*$_{OXA-1}$, *dfr*A19, *sul*1, *cat*B3, *ant*(2″)-Ia, *cat*A1, *tet*(B), *qnr*B34, *bla*$_{CMY-98}$ | TC, STR, CL, AMP, CI, CN, CTX | |
| | pCIW5.1_1 | | Col(pHAD28), Col440II | 5,410 | 52.26 | | | mobilizable |
| | pCIW5.1_2 | | | 2,715 | 32.41 | | | non-mobilizable |
| Ciw5.2 | | 4,812 | | 5,076,889 | 52.08 | *aad*A1, *bla*$_{OXA-1}$, *cat*A1, *tet*(B), *qnr*B34, *bla*$_{CMY-98}$ | TC, STR, CL, AMP, CI, CTX | |
| | pCIW5.2_1 | | Col440II, Col(pHAD28) | 5,410 | 52.26 | | | Mobilizable |
| | pCIW5.2_2 | | | 2,715 | 32.41 | | | Non-mobilizable |
| Cipa6.1 | | 4,374 | | 4,696,996 | 51.63 | | AMP | |
| Cipa6.2 | | 4,371 | | 4,695,525 | 51.63 | | (AMP) | |
| Cif11 | | 4,849 | | 4,921,369 | 51.74 | *bla*$_{CMY-48}$ | AMP | |
| | pCIF11_1 | | pKPC-CAV1321 [Col(VCM04)] | 228,564 | 47.47 | (Mercury resistance genes) | | Mobilizable |
| Cipo13 | | 5,182 | | 4,882,283 | 52.03 | *qnr*B9, *qnr*B7, *bla*$_{CMY-34}$ | TC, STR, AMP | |
| | pCIPO13_1 | | IncHI1A, IncHI1B, (IncHI1B) | 326,286 | 47.14 | (Tellurium resistance genes) | | Conjugative (MPF$_F$) |
| | pCIPO13_2 | | IncFIB, (IncFIB) | 108,873 | 50.50 | | | Non-mobilizable |
| | pCIPO13_3 | | IncFII(Yp), IncFIB(K), (IncFIB, IncFII) | 91,270 | 53.01 | *sul*2, *tet*(D) | | Conjugative (MPF$_T$) |
| | pCIPO13_4 | | | 39,641 | 48.54 | | | Mobilizable |
| | pCIPO13_5 | | IncN2, (IncN) | 32,965 | 50.35 | | | Conjugative (MPF$_T$) |
| Endy1 | | 4,237 | | 4,554,876 | 55.85 | *bla*$_{ACT-6}$, *fos*A | AMP, (CI) | |
| Endy2 | | 4,237 | | 4,554,873 | 55.84 | *bla*$_{ACT-6}$, *fos*A | AMP, (CI) | |
| Enh11 | | 4,531 | | 4,701,271 | 55.70 | *bla*$_{ACT-7}$, *fos*A | AMP, (CI) | |
| | pENH11_1 | | IncFIB, IncFII, (IncFIB, IncFII) | 158,457 | 52.42 | | | Conjugative (MPF$_F$) |
| Enb12 | | 4,606 | | 4,836,219 | 56.07 | *fos*A | AMP | |
| | pENB12_1 | | | 82,766 | 46.89 | | | Conjugative (MPF$_F$) |
| | MStein-2023a | | | 45,550 | 50.77 | | | Non-mobilizable |
| Ec1115 | | 4,917 | | 4,859,272 | 50.88 | | TC, STR, (AMP) | |
| | pEC1115_1 | | IncFIB, IncQ1, IncFII, (IncFIA, IncFIB, IncFIC, IncQ1) | 179,575 | 50.62 | *tet*(A), *dfr*A5, *bla*$_{TEM-1B}$, *sul*2, *aph*(3″)-Ib, *aph*(6)-Id (mercury resistance genes) | | Conjugative (MPF$_F$) |
| | pEC1115_2 | | IncI1-I, (IncI-gamma/K1) | 92,661 | 49.88 | | | Conjugative (MPF$_I$) |
| | pEC1115_3 | | IncFII, (IncFIA, IncFII) | 68,264 | 51.81 | | | Conjugative (MPF$_F$) |
| | pEC1115_4 | | | 3,374 | 55.19 | | | Mobilizable |
| Ec1117 | | 4,810 | | 4,933,597 | 50.80 | | TC, STR, AMP | |

*(Continued on next page)*

**TABLE 2** Results of bioinformatic analyses (e.g., ResFinder, PlasmidFinder, and Mob-suite) of complete chromosome and plasmid sequences of all strains relating to genome characteristics, antibiotic resistance genes, heavy metal resistance genes, and predicted mobility (*Continued*)

| Strain name/ chrom. | Plasmid | Total CDSs | PlasmidFinder (Mob-suite rep-type) | Sequence length (bp) | mol% GC content | Resistance genes (metal resistance genes) | Phenotypic resistance | Predicted mobility (mpf type) |
|---|---|---|---|---|---|---|---|---|
| | pEC1117_1 | | IncFIB, (IncFIB) | 119,797 | 50.64 | *tet*(A), *dfr*A5 | | Conjugative (MPF$_F$) |
| Ec1119 | | 4,501 | | 4,750,609 | 50.79 | | AMP | |
| | pEC1119_1 | | (IncN) | 34,007 | 48.71 | | | Conjugative |
| Ec1120 | | 4,446 | | 4,577,716 | 50.88 | | TC | |
| | pEC1120_1 | | p0111, (IncHI1B) | 90,204 | 48.92 | *tet*(A) | | Non-mobilizable |
| Kgr1 | | 5,433 | | 5,850,048 | 55.89 | *oqx*B, *bla*$_{OXY-6-4}$ | TC, AMP, CI | |
| | pKGR1_1 | | Col440II, Col(pHAD28) | 5,665 | 48.70 | | | Mobilizable |
| | pKGR1_2 | | | 4,721 | 52.09 | | | Mobilizable |
| | pKGR1_3 | | Col440I, (ColRNAI) | 4,096 | 55.52 | | | Mobilizable |
| | pKGR1_4 | | | 3,173 | 48.00 | | | Non-mobilizable |
| Kva3 | | 5,243 | | 5,517,735 | 57.33 | *oqx*B, *oqx*A, *bla*$_{LEN7}$, *fos*A | AMP | |
| Kpneu4 | | 5,347 | | 5,315,652 | 57.33 | *oqx*B, *oqx*A, *bla*$_{SHV-26}$, *fos*A | CL, AMP, CI | |
| | pKPNEU4_1 | | IncFIB(K), IncFIA, (IncFIA, IncFIB) | 234,572 | 51.10 | (Tellurium resistance genes) | | Non-mobilizable |
| | pKPNEU4_2 | | Col440I, (ColRNAI rep_cluster_1987, ColRNAI rep_clus-ter_1987) | 6,455 | 54.64 | | | Non-mobilizable |
| Kpneu8 | | 5,192 | | 5,349,345 | 57.38 | *oqx*B, *oqx*A, *bla*$_{SHV-27}$, *fos*A | TC, (CL), AMP, CI | |
| | pKPNEU8_1 | | IncR, (IncR) | 51,693 | 54.13 | *dfr*A14, *qnr*S1, *qnr*S1, *bla*$_{LAP-2}$, *tet*(A), *sul*2 | | Non-mobilizable |
| | pKPNEU8_2 | | Col440I | 2,987 | 45.36 | | | Non-mobilizable |
| Kpneu28 | | 5,364 | | 5,264,246 | 57.49 | *oqx*B, *oqx*A, *bla*$_{SHV-40}$, *fos*A | TC, AMP, CI, CTX | |
| | pKPNEU28_1 | | IncFIB(K), (IncFIB) | 139,211 | 51.66 | (Copper/silver resistance genes, arsenic resistance genes) | | Non-mobilizable |
| | pKPNEU28_2 | | IncR, IncFII(K), (IncFII, IncR) | 121,449 | 51.76 | *sul*1, *dfr*A1, *tet*(A), *qnr*S1, *bla*$_{TEM-1B}$, *bla*$_{CTX-M-15}$ | | Conjugative (MPF$_F$) |
| | pKPNEU28_3 | | | 42,330 | 50.34 | | | Conjugative (MPF$_T$) |
| | pKPNEU28_4 | | Col440I | 4,686 | 44.15 | | | Non-mobilizable |
| Kpneu34 | | 5,282 | | 5,262,405 | 57.48 | *oqx*B, *oqx*A, *bla*$_{SHV-28}$, *fos*A | TC, STR, CL, AMP, CI, CN, CTX | |
| | pKPNEU34_1 | | IncFIB(K), IncFII(K), (IncFIB, IncFII) | 209,900 | 53.29 | *aph*(3′)-Ia, *mph*(A), *sul*1, *aad*A2, *dfr*A12, *cat*A1 (copper/silver resistance genes, arsenic resistance genes) | | Conjugative (MPF$_F$) |
| | pKPNEU34_2 | | IncR, IncFIA(HI1), (IncFIA, IncR) | 63,972 | 52.71 | *aac* (3)-IId, *bla*$_{SHV-2}$, *tet*(D), *sul*1, *aad*A16, *dfr*A27, *arr*-3, *aac*(6′)-Ib-cr, *sul*2, *aph*(3″)-Ib, *aph* (6)-Id, *bla*$_{TEM-1B}$ (mercury resistance genes) | | Mobilizable |
| | pKPNEU34_3 | | Col(pHAD28), (ColRNAI rep_cluster_1987) | 4,167 | 41.49 | | | Non-mobilizable |
| | pKPNEU34_4 | | (ColpVC) | 2,058 | 56.51 | | | Non-mobilizable |

**TABLE 3** Minimum inhibitory concentration (MIC; in µg/mL) of the 22 enterobacterial strains against antibiotics[a]

| Strain | TC | STR | CL | AMP | CI | CN | CTX |
| --- | --- | --- | --- | --- | --- | --- | --- |
| | Tetracyclines | Aminoglycoside | Phenicols | Penicillin | Quinolones | Aminoglycoside | Cephems |
| Cigi1 | 256 | >256 | 16 | 32 | ≤0.25 | 2 | ≤0.5 |
| Cipo4 | 256 | 64 | 8 | 32 | ≤0.25 | 2 | ≤0.5 |
| Ciw5.1 | 256 | 128 | >256 | >256 | 16 | 32 | 8 |
| Ciw5.2 | 256 | 128 | >256 | >256 | 8 | 1 | 32 |
| Cipa6.1 | 2 | 4 | 8 | 64 | ≤0.25 | 1 | ≤0.5 |
| Cipa6.2 | 2 | 4 | 8 | 16 | ≤0.25 | ≤0.5 | ≤0.5 |
| Cif11 | 2 | 16 | 8 | 256 | ≤0.25 | 1 | 1 |
| Cipo13 | 256 | 32 | 8 | 64 | ≤0.25 | 1 | ≤0.5 |
| Endy1 | 4 | 8 | 8 | 256 | 0.5 | 1 | ≤0.5 |
| Endy2 | 4 | 4 | 4 | 128 | 0.5 | 1 | ≤0.5 |
| Enh11 | 4 | 8 | 8 | >256 | 0.5 | 1 | ≤0.5 |
| Enb12 | 4 | 8 | 8 | >256 | ≤0.25 | 1 | ≤0.5 |
| Ec1115 | 64 | >256 | 4 | 16 | ≤0.25 | 4 | ≤0.5 |
| Ec1117 | 64 | >256 | 8 | >256 | ≤0.25 | 1 | ≤0.5 |
| Ec1119 | 4 | 16 | 8 | >256 | ≤0.25 | 4 | ≤0.5 |
| Ec1120 | 256 | 16 | 8 | 8 | ≤0.25 | 2 | ≤0.5 |
| Kgr1 | 128 | 4 | 2 | 64 | 1 | ≤0.5 | ≤0.5 |
| Kva3 | 2 | 4 | 4 | 32 | ≤0.25 | 1 | ≤0.5 |
| Kpneu4 | 2 | 8 | 32 | 256 | 1 | 1 | ≤0.5 |
| Kpneu8 | 128 | 2 | 16 | 256 | 32 | 1 | ≤0.5 |
| Kpneu28 | 128 | 4 | 4 | >256 | 1 | 1 | >256 |
| Kpneu34 | >256 | >256 | >256 | >256 | 64 | 256 | 4 |

[a]TC, tetracycline; STR, streptomycin; CL, chloramphenicol; AMP, ampicillin; CI, ciprofloxacin; CN, gentamicin; CTX, cefotaxime. Breakpoints: tetracycline ($s ≤ 4$ µg/mL; $r ≥$ 16 µg/mL), streptomycin ($s ≤ 16$ µg/mL; $r ≥ 32$ µg/mL), chloramphenicol ($s ≤ 8$ µg/mL; $r ≥ 32$), ampicillin ($s ≤ 8$ µg/mL; $r ≥ 32$ µg/mL), ciprofloxacin ($s ≤ 0.25$ µg/mL; $r ≥$ 1 µg/mL), gentamicin ($s ≤ 4$ µg/mL; $r ≥ 16$ µg/mL), and cefotaxime ($s ≤ 1$ µg/mL; $r ≥ 4$ µg/mL). Strains were considered intermediate resistant when resistance values were higher than breakpoint indicating susceptibility, or lower than breakpoint indicating resistance. Breakpoints were used as suggested by CLSI (1) and for streptomycin by the US Food and Drug Administration (2).

the 22 strains (50%) showed multiple resistance to at least three different classes of antibiotics (mainly against tetracyclines, aminoglycosides, and β-lactams), while three strains showed resistance to at least six antibiotics (tetracycline, streptomycin, chloramphenicol, ampicillin, ciprofloxacin, and cefotaxime). Only two strains, *K. pneumoniae* strain Kpneu34 and *Citrobacter werkmanii* strain Ciw5.1, showed resistance to gentamicin (Table 3).

## Genomic characteristics

The largest contig obtained from each strain of genomic sequencing represented the complete bacterial chromosomal DNA (>4.5 Mbp in all cases). The complete DNA sequences of 22 *Enterobacteriaceae* strains were used to identify the bacteria-to-species level by dDDH (Table 1), and the *Citrobacter* strains were found to include the species *C. gillenii* ($n = 1$), *C. portucalensis* ($n = 2$), *C. werkmanii* ($n = 2$), *Citrobacter pasteurii* ($n = 2$), and *Citrobacter freundii* ($n = 1$); the *Enterobacter* strains included the species *En. dykesii* ($n = 2$), *Enterobacter hormaechei* ($n = 1$), and *Enterobacter bugandensis* ($n = 1$); and the *Klebsiella* strains included the species *Klebsiella grimontii* ($n = 1$), *Klebsiella variicola* ($n = 1$), and *K. pneumoniae* ($n = 4$), while the *Escherichia* strains included only the species *E. coli* ($n = 4$).

The genome sizes of the strains in this study varied from 4.55 to 5.85 Mbp, with *K. grimontii* exhibiting the largest and the two *En. dykesii* strains having the smallest genomes of all investigated strains (Table 2). When considering the *Citrobacter* spp., the chromosomal DNA sizes ranged from 4.69 to 5.09 Mbp, with the two *C. pasteurii* strains having the smallest genomes of 4.69 Mbp. The *Enterobacter* spp. genome sizes ranged from 4.55 Mbp (*En. dykesii*) to 4.84 Mbp (*En. bugandensis*), while those for *E. coli* strains ranged from 4.58 to 4.93 Mbp. Overall, the *Klebsiella* genomes were the largest chromosomes in this study ranging from 5.26 Mbp (*K. pneumoniae*) to 5.85 Mbp (*K.*

*grimontii*) (Table 2). The chromosomal mol% GC contents ranged from 50.79% to 57.49% (differing by a maximum of 6.7 mol%), with *E. coli* Ec1119 having the lowest and *K. pneumoniae* Kpneu28 having the highest mol% GC content. The mol% GC content varied noticeably between genera but was similar within a genus. *Escherichia* and *Citrobacter* had rather low mol% GC contents ranging from 50.79% to 50.88% and 51.63% to 52.5%, respectively, while *Enterobacter* and *Klebsiella* showed higher mol% GC contents ranging from 55.7% to 56.07% and 55.89% to 57.49%, respectively.

For each of the three pairs of strains showing high similarity in chromosomal DNA size (Endy1 and Endy2, Cipa6.1 and Cipa6.2, and Ciw5.1 and Ciw5.2), the dDDH analysis resulted in 100% identity, despite length differences of 3, 1,471, and 8,809 bp, respectively. The strains Endy1 and Endy2, as well as Cipa6.1 and Cipa6.2, showed similar phenotypic antibiotic resistance patterns, were isolated from the same fresh produce (Tables 1 and 3), and were identified as clonal isolates. In our previous study, there were also some differences between these isolates in their phenotypic characteristics, including sugar fermentation tests (18). Strains Ciw5.1 and Ciw5.2 were isolated from the same product, i.e., China rose sprouts (Table 1). The nucleotide differences between Ciw5.1 (5.08 Mbp) and Ciw5.2 (5.07 Mbp) occurred at different chromosomal locations, of which the largest coherent differing sequence of 8,152 bp was inspected more closely (Fig. 1). The data showed that there was an insertion/deletion of a resistance region, including an integron. Strain Ciw5.2 possessed one class 1 integrase, while strain Ciw5.1 possessed two class 1 integrases and nine additional genes in between those two integrases.

Five of the 22 strains (Cipa6.1, Cipa6.2, Endy1, Endy2, and Kva3) did not harbor any extrachromosomal DNA sequences. The remaining 17 strains (77%) harbored in total 41 extrachromosomal DNA sequences (including two identical sequences: pCIW5.1_1 and pCIW5.2; pCIW5.1_2 and pCIW5.2_2), ranging from 2,058 to 326,286 bp (Table 2). One of the extrachromosomal DNA sequences of the strain Enb12 was identified as a

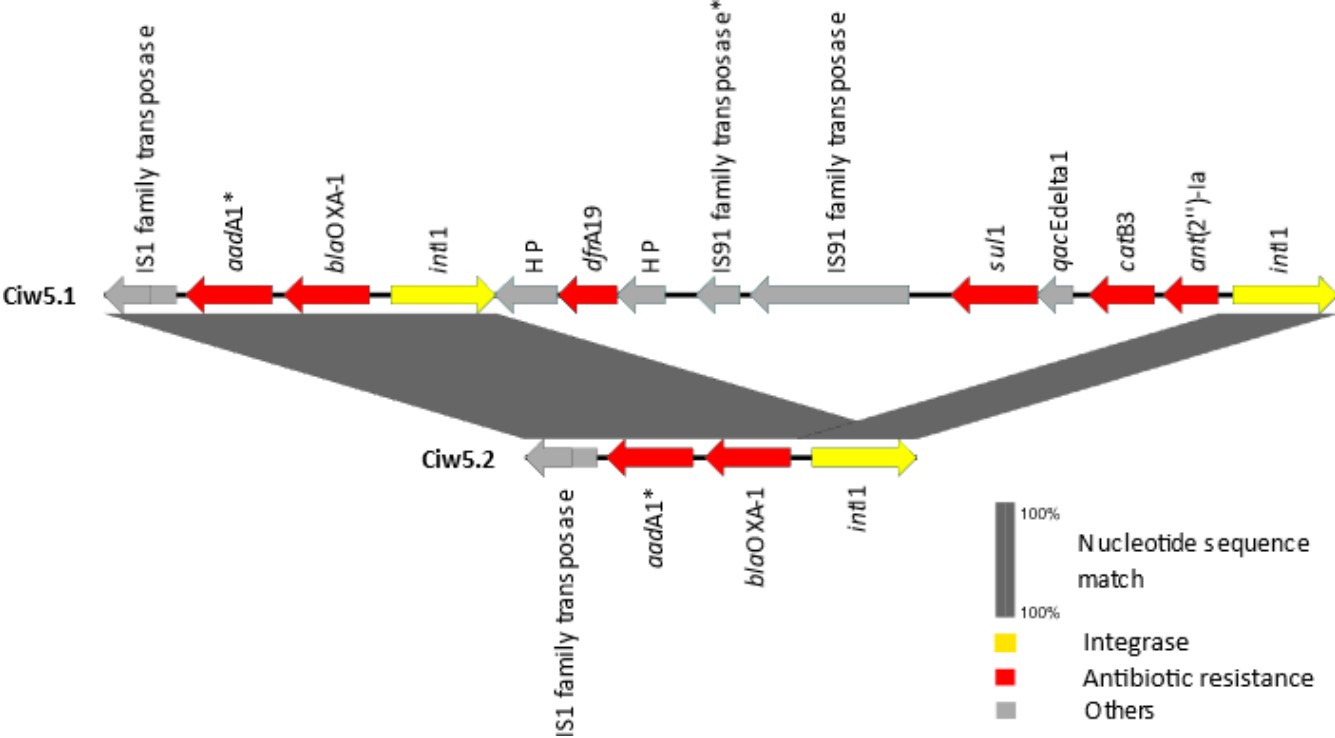

**FIG 1** Comparisons of corresponding chromosomal loci of strains Ciw5.1 (nucleotides 67,740 to 76,677 bp) and Ciw5.2 (nucleotides 64,740 to 68,530 bp) using EasyFig and Prokaryotic Genome Annotation Pipeline (PGAP) annotation. In the case gene names/descriptors were not available, the gene products of the coding sequences are indicated. *Identified by PGAP annotation as a pseudogene.

self-replicating prophage using PHASTER (PHAge Search Tool Enhanced Release) (22, 23). Most of this 45,550-bp sequence was classified as an intact prophage (score = 105) with 58 protein-coding sequences. This Enb12 DNA sequence showed the highest similarity to prophage N15 in the blastn analysis. This prophage N15 was characterized as an extrachromosomal, linear prophage (24, 25). The Enb12 prophage was named MStein-2023a upon submission of the sequence to the NCBI GenBank database and was excluded from further description of plasmids in this study (Table 2).

The 38 unique plasmids detected in the 17 strains occurred at numbers between one and five different plasmids per strain. Mostly, strains harbored one (six strains) or two (five strains) plasmids (Table 2). In this study, approx. one-third (14 plasmids, 36.8%) of plasmids were considered small plasmids (defined here as <7,000 bp), while the remaining approx. two-thirds (24 plasmids, 63.1%) were considered large plasmids (defined here as >32,000 bp). In total, 28.9% (11/38) of the plasmids carried antibiotic resistance genes [i.e., *tet*(A) and *sul*2). It was noted that none of the small plasmids carried such genes.

## AMR

Most *Citrobacter* strains, except for the strains Cigi1, Cipa6.1, and Cipa6.2, possessed resistance genes on their chromosome. Notably, *C. werkmanii* strain Ciw5.1 possessed the most diverse antibiotic resistance genes on chromosomal DNA (10 genes conferring resistance to seven antibiotic classes) (Fig. 1; Table 2). *Enterobacter* spp. and *Klebsiella* spp. showed similar resistance gene patterns, with *fos*A and β-lactam resistance genes predominating on their chromosome, while *E. coli* strains did not exhibit resistance genes on their chromosome. Overall, the most prevalent antibiotic resistance genes on chromosomes of the 22 *Enterobacteriaceae* strains were quinolone, β-lactam, and fosfomycin resistance genes in this study (Tables 2 and 3).

Of 17 strains with a total of 38 unique plasmids found in this study, only nine strains harbored plasmids with resistance genes. The strain Cipo4 harbored two plasmids, which both encoded resistance genes, and Kpneu34 possessed four plasmids of which two encoded resistance genes. None of the small plasmids (<7,000 bp) carried resistance genes, and therefore, only large plasmids of between 49,031 and 209,900 bp harbored resistance genes. The largest plasmid found in this study (pCIPO13_1; 326,286 bp), belonging to the IncHI1A(NDM-CIT) and IncHI1B (pNDM-CIT) incompatibility groups, did also not carry any antibiotic resistance genes (Table 2). Among the large plasmids carrying antibiotic resistance genes, there was no noticeable relationship between plasmid size and the number of antibiotic resistance genes, since, e.g., the plasmid with the most resistance genes (pKPNEU34_2) was only 63,972 bp in size while the larger plasmid pEC1117_1 (119,797 bp) carried only two resistance genes. None of the *Enterobacter* plasmids (0/2), 25% of the *Klebsiella* plasmids (4/16), 30% of the *Citrobacter* plasmids (4/13), and 42% of the *E. coli* plasmids (3/7) harbored antibiotic resistance genes. Plasmids carried either 1, 2, 3, 6, or 12 antibiotic resistance genes. The plasmid maps of pEC1115_1 and pKPNEU28_2 show the antibiotic resistance genes located on these plasmids (Fig. 2A and B).

The most prevalent antibiotic resistance genes on plasmids were those encoding aminoglycoside (*n* = 11 genes), tetracycline (*n* = 10 genes), and sulfonamide (*n* = 7 genes) resistance. Except for one antibiotic resistance plasmid, all carried tetracycline resistance genes *tet*(A), *tet*(D), and *tet*(Y), among which *tet*(A) was the most prevalent. For plasmids pCIPO4_1, pCIPO4_2, and pEC1120_1, tetracycline resistance genes were the only resistance determinants present on these plasmids. *Citrobacter* plasmids had a maximum of three antibiotic resistance genes, against tetracycline and either aminoglycoside or sulfonamide antibiotics. *Escherichia coli* plasmid pEC1119_1 had no resistance genes, and pEC1120_1 had *tet*(A), while pEC1117_1 had *tet*(A) and *dfr*A as resistance determinants. The Ec1115 strain, on the other hand, harbored four plasmids, but only the largest plasmid contained six antibiotic resistance genes [*aph* (6)-Id*, aph*(3″)-Ib, *tet*(A), *dfr*A5, *bla*_TEM-1B, and *sul*2] (Table 2). *Klebsiella pneumoniae* resistance plasmids

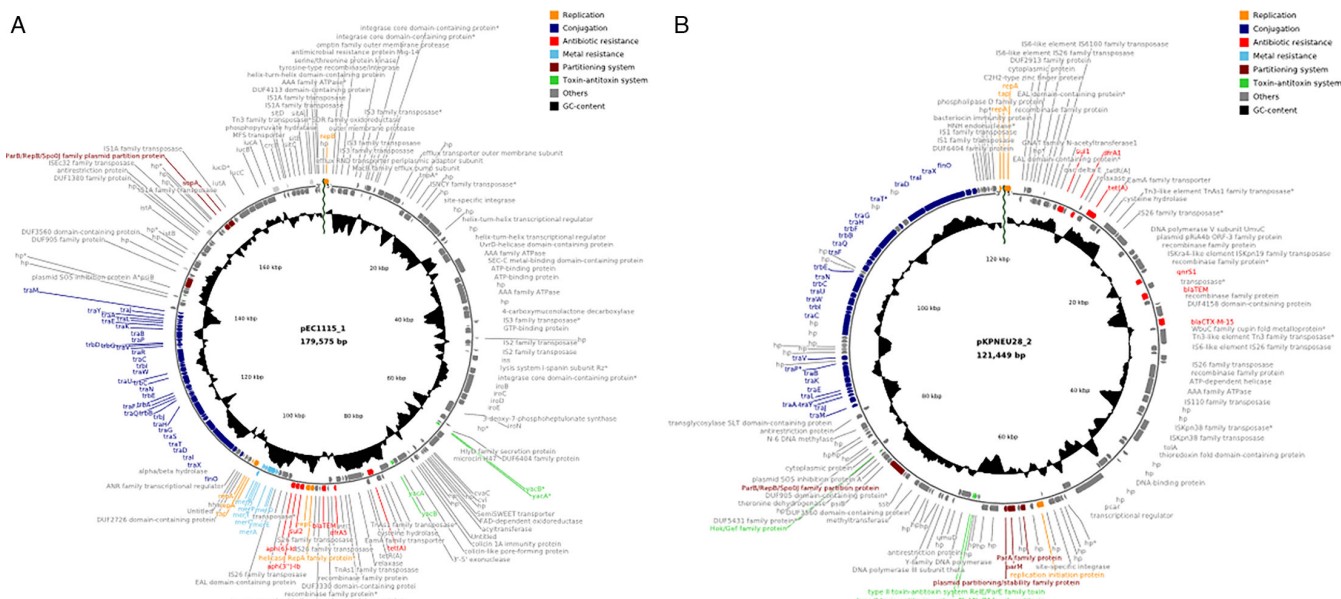

**FIG 2** Plasmid maps of the antibiotic resistance plasmid pEC1115_1 from the *E. coli* strain Ec1115 (A) and the antibiotic resistance plasmid pKPNEU28_2 from the *K. pneumoniae* strain Kpneu28 (B). The plasmid maps were prepared using CGview on the basis of PGAP annotation data. In the case gene names/descriptors were not available, the gene products of the coding sequences are indicated. HP, hypothetical protein. *indicated by PGAP to represent a pseudogene.

harbored 6 or 12 antibiotic resistance genes against at least four antibiotic classes (Table 2). The resistance plasmids of Kpneu34 each had two resistance genes that were unique for plasmids in this study. On pKPNEU34_1, a macrolide resistance gene *mph*(A) and a phenicol resistance gene *cat*A1 were detected. Plasmid pKPNEU34_2 possessed among other resistance genes the rifampicin resistance gene *arr*-3 and the aminoglycoside acetyltransferase gene *aac*(6′)-Ib-cr. The plasmid pKPNEU34_2 possessed the most antibiotic resistance genes (*n* = 12) that encoded resistance to six different classes of antibiotics. The two phenotypically most resistant bacterial strains Kpneu34 and Ciw5.1 were isolated from sprouts and possessed four and two plasmids, respectively. Interestingly, the *K. pneumoniae* Kpneu34 strain had most of its resistance genes on the two large plasmids and only few resistance genes on the chromosome, while the Ciw5.1 and the Ciw5.2 strains had all resistance genes only on the chromosome (Table 2).

## Plasmids/transposons/integrons

PlasmidFinder identified 11 different plasmid replicon types, including IncFIB (*n* = 9), IncFII (*n* = 6), IncFIA (*n* = 2), IncI (*n* = 1), IncR (*n* = 4), IncHI1A/B (*n* = 1), IncQ (*n* = 1), IncN (*n* = 1), Col440I (*n* = 4), Col440II (*n* = 3), and Col(pHAD28) (*n* = 3) (Table 2). For 13 sequences identified as complete extrachromosomal sequences, there was no match with a known replicon type in the PlasmidFinder database. A single replicon type was identified in 16 plasmids, while in nine plasmids, two, and in one plasmid, even three replicon types were detected. IncFIB was the most prevalent, followed by IncFII. In some plasmids, both replicons were located on the same plasmid and also in combination with replicon types IncQ1 or IncR. In small plasmids, only Col-plasmids [Col440I, Col440II, and Col(pHAD28)] were detected, while Col-like sequences were not identified in any of the large plasmids in this study. However, the Mob-suite pipeline identified a Col-replicon type on a single large plasmid (pCFR11_1). The Col440II and Col(pHAD28) replicon types were almost always found occurring together on different plasmids (pCIW5.1_1 and pKGR1_1; see Table 2). All four IncR plasmids found in this study possessed antibiotic resistance genes, while only some of the IncFII plasmids did, and others did not possess any antibiotic resistance genes. No resistance genes were detected on plasmids with IncN and IncI replicons. The results obtained using

the Mob-suite pipeline were similar to those of PlasmidFinder, but not always identical. IncFIB, IncFII, IncI, and IncR were classified identically, whereas, IncFIA, IncHI1B, IncN, and Col replicon types were identified in some strains where PlasmidFinder failed to identify these replicon types (Table 2). For plasmids pCIF11_1, pEC1119_1, pEC1120_1, and pKPNEU34_4, the replicon types Col (VCM04), IncN, IncHI1B, and ColpVC were assigned by the Mob-suite pipeline, but not by PlasmidFinder. Furthermore, Mob-suite could predict the mobility of plasmids. Among the plasmids in this study, 36.8% were predicted to be conjugative, 28.9% were mobilizable, and 36.8% were non-mobilizable (Table 2).

Besides antibiotic resistance genes, heavy metal resistance genes were also detected on plasmid sequences in this study (Table 2). Similar to antibiotic resistance genes, heavy metal resistance genes were only detected on large plasmids. The smallest plasmid that possessed heavy metal resistance genes was pKPNEU34_2 (63,972 bp) (Table 2). No heavy metal resistance genes were found on *Enterobacter* species plasmids in this study. For *E. coli* plasmids, only pEC1115_1 (Fig. 2A) harbored a mercury resistance operon. Genes potentially conferring resistance to tellurium, silver, copper, arsenic, and mercury were found on some *Klebsiella* plasmids. *Citrobacter* plasmids encoded genes for mercury, copper, and tellurium resistance (Table 2). Mercury resistance genes were the most prevalent heavy metal resistance genes on plasmids in this study. Overall 7 of the 38 plasmids (18.4%) carried heavy metal resistance genes (Table 2).

Using the MobileElementFinder tool, composite transposons and unit transposons were identified on the plasmids (Table 4). Transposons carried genes for semi-metal and heavy metal resistances (pCIF11_1, pKPNEU4_1, pKPNEU28_1, and pKPEU34_1), disinfectant resistance (pKPNEU28_2 and pKPNEU34_2), toxin–antitoxin systems (pCIPO4_2, pCIF11_1, pCIPO13_1, pCIPO13_3, pKPNEU4_1, pKPNEU28_1, and pKPNEU34_1), and replication proteins [pCIPO4_2 (IncR), pEC1115_1 (IncQ), and pKPNEU34_2 (IncFIA)] (Table 4). Different insertion sequences were found to be associated with the composite transposons, but interestingly, IS26 was found to be part of all composite transposons that included antibiotic resistance genes. On *Enterobacter* plasmids, no antibiotic resistance genes and no composite transposons or unit transposons could be detected. On plasmids in this study, 50% of antibiotic resistance genes were associated with composite transposons and unit transposons.

The Integron Finder identified complete class1 integrons with the respective cassettes on plasmids, pEC1115_1, pEC1117_1, and pKPNEU34_1, as well as on the chromosome of Ciw5.1 (two complete integrons) and Ciw5.2 (Fig. 3). The plasmid pKPNEU34_2 harbored an incomplete integron, where the gene cassettes were present, but the integrase was

**TABLE 4** Antibiotic resistance genes with mobile genetic elements and plasmid typical genes[a,b]

| Plasmid | Type of MGE | Position sequence | AMR genes | Additional genes |
|---|---|---|---|---|
| pCIPO4_1 | Composite transposase, cn_5166_IS26 | 7,798–12,964 | *tet*(D) | |
| pCIPO13_3 | Composite transposase, cn_33404_IS26 | 14,640–48,044 | *sul*2 | Type II toxin and antitoxin toxin (RelE/ParE) |
| | Composite transposase, cn_12876_IS5075 | 40,843–53,719 | *sul*2, *tet*(D) | |
| | Composite transposase, cn_5163_IS26 | 47,224–52,387 | *tet*(D) | |
| pEC1115_1 | Composite transposase, cn_3556_IS26 | 88,051–91,607 | *bla*$_{TEM-1B}$ | |
| | Composite transposase, cn_6026_IS26 | 90,787–96,813 | *aph*(6)-Id, *aph*(3″)-Ib, *sul*2 | RepC (IncQ) |
| pEC1120_1 | Unit transposon, Tn1721 | 56,428–67,555 | *tet*(A) | |
| pKPNEU28_2 | Composite transposase, cn_11923_IS26 | 3,452–15,375 | *tet*(A), *sul*1, *dfr*A1 | *qacE* |
| pKPNEU34_2 | Composite transposase, cn_6367_IS26 | 900–7,267 | *aac*(3)-IId | |
| | Composite transposase, cn_5561_IS26 | 6,447–12,008 | *bla*$_{SHV-2}$ | |
| | Composite transposase, cn_9994_IS26 | 11,188–21,182 | *tet*(D) | RepE [FIA (HI1)] |
| | Composite transposase, cn_11896_IS26 | 20,362–32,258 | *sul*1, *aad*A16, *dfr*A27, *arr*-3, *aac*(6′)-Ib-cr | *qacE* |
| | Composite transposase, cn_9271_IS26 | 31,438–40,709 | *sul*2, *aph*(3″)-Ib, *aph* (6)-Id, *bla*$_{TEM-1B}$ | |

[a](26).
[b](27).

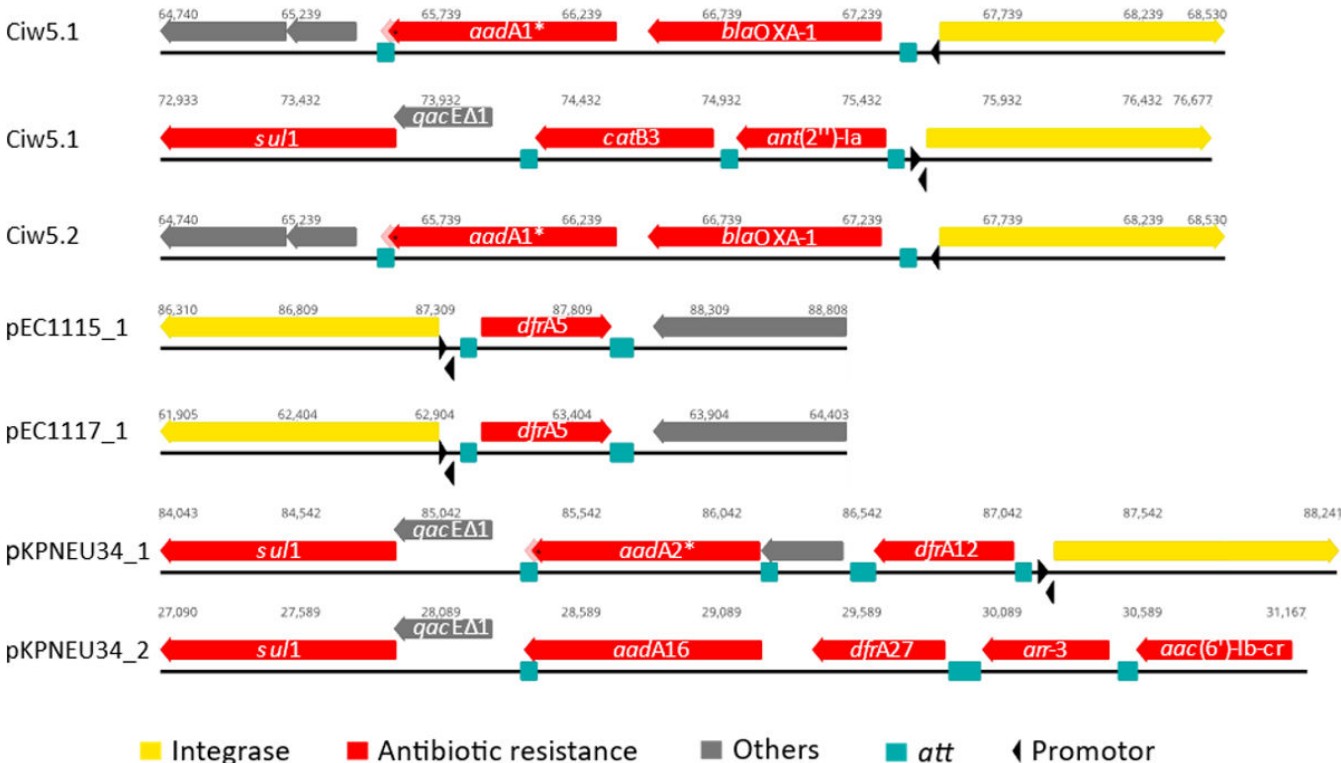

FIG 3  Class 1 integrons detected on chromosome and plasmid sequences of antibiotic-resistant enterobacteria from fresh produce using IntegronFinder. Only a partial integron was found on plasmid pKPNEU34_2 as a corresponding integron integrase could not be identified from the sequence. *Identified by PGAP annotation as a pseudogene.

missing. In all integron cassettes, antibiotic resistance genes were included. The cassettes of pEC1115_1 and pEC1117_1 were identical and contained the IS26 insertion sequence. Chromosomes of Ciw5.1 and Ciw5.2 shared an identical insertion sequence, while Ciw5.1 had an additional cassette, which Ciw5.2 did not possess (Fig. 3).

## DISCUSSION

Eighteen out of 22 strains in this study were previously sequenced using MiSeq (18). However, it was not possible to complete the sequences of chromosomes and plasmids without sequence gaps solely using MiSeq sequencing data in the previous study. Plasmids are still challenging to assemble into one entire sequence, as repeat elements often do not allow the resolution of the complete sequences and, therefore, generate numerous fragmented contigs (28). The combination of short-read and long-read sequencing data in this study was successfully used to generate complete genome sequences, including extrachromosomal DNA such as plasmids and prophage. Additionally, these sequences were identified as circular by the Unicycler pipeline and were considered complete (29). Thus, it was possible to fully resolve the chromosomes of 22 *Enterobacteriaceae* strains, as well as all of the 38 unique plasmids and one extrachromosomal prophage harbored by the strains in this study.

Strain identification was accomplished using dDDH with complete genome sequence data. The results obtained confirmed the results of the previous study (18), in which the MiSeq data of all strains (except for strains Cif11, Kva3, Kpneu8, Kpneu28, and Kpneu34, which were determined in this study) were used to identify strains using different genotyping and whole genome analysis approaches (Table 1). Based on dDDH analysis with the reference strain *C. gillenii* AF64_5pH9A, strain Cigi1 could be identified as a *C. gillenii* strain in this study.

In previous years, generally high antibiotic resistance incidences of *Klebsiella, Enterobacter,* and *Escherichia* spp. isolated from fresh produce have been reported (30–33). Strains were described to possess phenotypic resistances to antibiotics such as ampicillin, tetracycline, and gentamicin (18, 19, 33). Many of the strains were also found to be multiple-resistant (30–33). According to CLSI guidelines, the ampicillin resistance of *K. pneumoniae*, *K. variicola,* and *C. freundii* can be explained by intrinsic resistance as a result of chromosomally encoded β-lactamase resistance genes (26). Such chromosomally encoded β-lactamase resistance genes, including *bla*$_{OXY}$, *bla*$_{LEN}$, *bla*$_{SHV}$, *bla*$_{ATC}$, *bla*$_{CMY}$, and *bla*$_{OXA}$, were indeed found on the chromosomes of *Citrobacter*, *Enterobacter,* and some *Klebsiella* strains in this study. Furthermore, it was noted that many of the *Enterobacteriaceae* isolates from fresh produce in this study contained fosfomycin resistance genes on their chromosomes, indicating also a possible intrinsic resistance to these antibiotics. In addition, almost half of the strains were susceptible to tetracycline. Most tetracycline resistance genes were previously described to be associated with mobile genetic elements (34). This was also observed in strains of this study, where *tet*(A), *tet*(D), and *tet*(Y) were associated with plasmids, whereas *tet*(B) was found on the chromosome of two *Citrobacter* strains. We were not able to find a resistance gene that was responsible for the observed tetracycline resistance in strain *K. grimontii* Kgr1. The most common resistance-coding genes on chromosomes differed greatly from the ones on the plasmids, as on these, the occurring resistance genes mostly included aminoglycosides, tetracycline, and sulfonamide, as well as diaminopyrimidines, quinolone, macrolide antibiotics, and chloramphenicol genes.

Plasmids are especially relevant for the horizontal gene transfer of antibiotic resistance (35, 36), and for this reason, this study focused on the complete sequencing of plasmids and identification of their resistance genes. Using both long- and short-read sequencing and a hybrid assembly, it was possible to generate the complete chromosomal sequences of all 22 strains as a single contig. Furthermore, the sequences of all plasmids present in the strains could be resolved. Whereas five strains did not contain any plasmids, the remaining 17 isolates together possessed 38 plasmids that differed in size, mobilization ability, presence of antibiotic resistance or heavy metal resistance genes, and mol% GC content. Considering the above characteristics such as size, mol% GC content, and types of genes present, clearly there is a highly diverse pool of potentially transferable DNA fragments in fresh produce, which obviously has a history of and potential for future horizontal dissemination.

Regarding plasmid transferability, it should be noted that the most common Inc types of the large plasmids occurring in the *Enterobacteriaceae* from fresh produce were IncF plasmids, especially IncFIB and IncFII. The smaller plasmids, when at all typable with PlasmidFinder, showed the highest similarities with Col replicon types. In a previous study by Blau et al. (19), the dominating replicon types of 63 *E. coli* strains isolated from fresh produce (coriander, mixed lettuce, and arugula) were also determined to be IncFIB and IncFII, while the IncI1 and IncX replicon types were also found to occur frequently. In this study, only one plasmid sequence of *E. coli* (pEC1115_2) had the IncI1 replicon type, in contrast to the study by Blau et al. (19). Furthermore, these complete plasmid sequences were utilized not only to classify the replication types of plasmids but also to analyze the entire set of gene features associated with plasmid functions, in comparison to the previous study (19). Huizinga et al. (37) investigated extended-spectrum β-lactamase-producing bacteria from sprouts in the Netherlands and also found Col and IncF replicon types to predominate in whole genome sequence data using PlasmidFinder (v. 2.1). They also identified the presence of replicons such as IncF, IncU, and IncN. Not all IncF plasmids in this study possessed antibiotic resistance genes. Nevertheless, 6 of the 11 plasmids carrying antibiotic resistance genes displayed the IncF replicon type or possessed multiple replicon types, with IncF being one of them. Five of these plasmids were determined to be potentially conjugative, which emphasizes the importance of this replicon type in the dissemination of antibiotic resistance genes. IncF plasmids indeed play an important role in the dissemination of resistance genes within

the *Enterobacteriaceae* (38, 39), and IncF plasmids have been found in *Enterobacteriaceae* from humans (38, 40, 41), from the environment (42, 43), in foods (42, 44), and in animals (41). This emphasizes the high potential and importance of these replicon types in the spread of antibiotic resistance genes not only in *Enterobacteriaceae* from fresh produce but possibly also from produce to *Enterobacteriaceae* in the human gut. IncF plasmids have been reported to contribute to the worldwide dissemination of clinically relevant antibiotic resistance genes, such as the β-lactamase $bla_{CTX-M-15}$ gene associated with an extended-spectrum β-lactamase activity (45, 46). Even in our study, such a $bla_{CTX-M-15}$ ESBL-associated gene could be detected on a plasmid from a *K. pneumoniae* strain that was isolated from sprouts.

There were also many replication proteins and, therefore, replicons for which PlasmidFinder was not able to find a matching known replicon type in the database. Notably, most of these non-assigned replicons occurred in *Citrobacter* strains. This might reflect the fact that *Citrobacter* appears to be less associated with nosocomial infections compared to the ESKAPE (*Enterococcus faecium*, *Staphylococcus aureus*, *Klebsiella pneumoniae*, *Acinetobacter baumannii*, *Pseudomonas aeruginosa*, and *Enterobacter* species) strains *Klebsiella* and *Enterobacter*, or the intensively studied *E. coli*. There, thus, might be a bias toward more widely studied plasmids from the genera *Klebsiella*, *Escherichia*, and *Enterobacter*, and consequently, less replicon typing sequences of *Citrobacter* plasmids may be available in the PlasmidFinder database for this reason. In this study, small plasmids were also found that were not at all typable by either PlasmidFinder or Mob-suite. Of all the small plasmids that could be typed, according to the replicon type, these were all assigned to the Col family. One of the large plasmids (pCIF11_1) was typed as a Col replicon by Mob-suite but not by PlasmidFinder. In this study, Mob-suite predicted 36.8% of plasmids being potentially conjugative; 28.9% were mobilizable (together 67%) while 36.8% were non-mobilizable. It was previously reported that approximately 54% of plasmids from Proteobacteria (*n* = 487) were conjugative and/or mobilizable, while 46% were non-mobilizable (47), which is relatively similar to the results of this study.

Overall, our study showed that *Enterobacteriaceae* from fresh produce possess both chromosomal and plasmid antibiotic resistance genes. While not all plasmids contained antibiotic resistance genes and not all plasmids were determined to be transferable by either conjugation or mobilization, indeed many of them were. Moreover, as some resistance genes were also found to be located on integrons possessing resistance genes within gene cassettes, there is a further dimension for the possible spread of these genes within the cell from chromosome to plasmids and then furthermore possibly also between cells. This indicates the potential for *Enterobacteriaceae* from fresh produce to contribute to the transfer of resistance genes either on such product or potentially also to *Enterobacteriaceae* of the gut if the produce is eaten raw and bacteria survive stomach and small intestinal transit. In this study, we focused on 12 species that belonged to *Enterobacteriaceae* from fresh produce in northern Germany and generated 22 complete genome sequences. However, the microbiota of fresh produce, including agricultural soil and irradiation water, are much more complex and also highly diverse. Furthermore, while we conducted complete genome sequences, we did not enumerate antibiotic-resistant bacteria present in the fresh produce. Therefore, the 12 species considered in this study may not provide sufficient data on the potential of antibiotic resistance plasmid spread from fresh produce, and more extensive research on this would be required.

## MATERIALS AND METHODS

### Bacterial strains and culturing conditions

Most of the strains used in this study were previously characterized and identified (18). The strain designation in the publication of Cho et al. (18) was amended to indicate the

name and the species of the strains as abbreviations. For comparability, the previous designations, where applicable, were included in parenthesis behind the current strain designations. Thus, the 22 strains Cigi1 (C1), Cipo4 (C4), Ciw5.1 (C5.1), Ciw5.2 (C5.2), Cipa6.1 (C6.1), Cipa6.2 (C6.2), Cif11, Cipo13 (C13), Endy1 (E1), Endy2 (E2), Enh11 (E11), Enb12 (E12), Kgr1 (K1), Kva3, Kpneu4 (K4), Kpneu8, Kpneu28, Kpneu34, Ec1115, Ec1117, Ec1119, and Ec1120 belonged to the species *C. gillenii, C. portucalensis, C. werkmanii, C. pasteurii, C. freundii, En. dykesii, En. hormaechei, En. bugandensis, K. grimontii, K. variicola, K. pneumoniae,* and *E. coli* (Table 1). The strains Kpneu28 and Kpneu34, isolated from sprouts, were not characterized and published before. In order to link the designation of plasmid DNA to the respective bacterial host, plasmid designations included the host genus and species initials and were numbered according to size [e.g., pEC1115_1, largest plasmid DNA (_1) of *E. coli* strain 1115 (EC1115)].

Strains were isolated from fresh produce including mixed salads, arugula, sprouts, cucumber, carrots, and herbs such as marjoram and oregano purchased from German retail markets (18) (Table 1). All strains except for Kpneu28 and Kpneu34 were isolated on violet red bile dextrose agar (Merck, Darmstadt, Germany) containing tetracycline as previously described (18). The strain Kpneu28 was isolated on Brilliance extended-spectrum β-lactamase (ESBL) agar (Oxoid Ltd, Altrincham, United Kingdom) while the strain Kpneu34 was isolated on Brilliance carbapenem-resistant *Enterobacteriaceae* (CRE) agar (Oxoid, Altrincham, United Kingdom). Both strains were isolated from fresh packaged sprouts, and for this, 25 g of sprouts was aseptically placed in a sterile stomacher bag, and 225 mL of buffered peptone water (VWR, Darmstadt, Germany) was added. The sprout samples were homogenized in a stomacher (Seward, West Sussex, United Kingdom) at 200 rpm for 120 s at room temperature. For enrichment, the sample was incubated at 37°C for 24 h and again homogenized in a stomacher as done before. One milliliter of the sample was diluted 1:10 in Brilliant Green Bile Lactose Broth (Merck) and incubated for an additional 24 h at 41.5°C. The increase in temperature serves to reduce the accompanying bacterial microbiota. After incubation, 10 µL of the enriched sample was spread onto Brilliance ESBL agar or Brilliance CRE agar to detect ESBL-producing or carbapenem-resistant *Enterobacteriaceae* and incubated at 41.5°C for 24 h. Blue colonies indicative of carbapenem-resistant *Klebsiella, Enterobacter, Serratia,* and *Citrobacter* group microorganisms were selected and purified as described above. Further characterization using the AmpC and ESBL test D68C (MAST Diagnostica, Reinfeld, Germany) was done according to the manufacturer's instructions and confirmed the isolates as ESBL producers. All further culturing was done in lysogeny broth (LB) (Roth, Karlsruhe, Germany) without antibiotics in flasks at 37°C with shaking at 130 rpm overnight.

## Antibiotic susceptibility testing

The minimal inhibitory concentration (MIC) for seven antibiotics belonging to five antibiotic classes [tetracyclines: tetracycline hydrochloride (Sigma, Steinheim, Germany); aminoglycosides: streptomycin sulfate (AppliChem, Darmstadt, Germany) and gentamicin sulfate (AppliChem); phenicols: chloramphenicol (AppliChem); fluoroquinolones: ciprofloxacin hydrochloride monohydrate (Thermo Fisher Scientific, Wesel, Germany); and β-lactam antibiotics: ampicillin sodium salt (AppliChem) and cefotaxime sodium salt (Sigma-Aldrich, Taufkirchen, Germany)] was determined using the broth dilution method according to CLSI guidelines. Briefly, an overnight broth culture of the strain grown at 37°C was diluted in Müller–Hinton broth 2 (Merck) to a density of 0.5 McFarland, and then, 10 µL was inoculated into a twofold dilution series performed with different antibiotics in 96-well plates (Thermo Fisher Scientific). *E. coli* ATCC 25922 was used as a control strain. All tests were performed in duplicate, and plates were visually evaluated after 20 h of incubation at 35°C. The lowest concentration of antibiotics that inhibited bacterial growth was considered as MIC and compared to the CLSI standard to determine resistance (26). Since there are no CLSI breakpoints for streptomycin, the breakpoints implemented by the Food and Drug Administration were used (48).

## Genomic DNA isolation, long- and short-read sequencing, and hybrid sequence assembly

All strains except Cif11, Kva3, Kpneu8, Kpneu28, and Kpneu34 were previously sequenced in a taxonomic study using the MiSeq sequencing platform (Illumina, San Diego, USA), and sequences were published as assembled contigs (18). For strains Cif11, Kva3, and Kpneu8, 2 mL of a fresh overnight culture in LB broth (Roth) grown at 37°C was harvested by centrifugation (6,000 × $g$), and total genomic DNA was extracted using the PeqGold Bacterial DNA Mini Kit (Peqlab, Erlangen, Germany) following the manufacturer's instruction. MiSeq (Illumina) sequencing was performed as previously described (18). For strains Kpneu28 and Kpneu34, the strains were incubated overnight in tryptic soy broth (Roth) at 37°C, and total genomic DNA was extracted from a 1.8-mL volume using the Quick-DNA Fungal/Bacterial Miniprep Kit (Zymo Research, Irvine, USA). The DNA was prepared using the TruSeq Nano DNA LT Library Prep Kit (Illumina), and the DNA library was paired-end-sequenced using the MiSeq Reagent Kit v2 (Illumina) on a MiSeq sequencing platform (Illumina). Trimming of MiSeq sequencing data was done using Trimmomatic (0.39) (27). Only paired reads were used for hybrid assembly.

For long-read sequencing of all strains in this study, a single colony of each strain was propagated in 20-mL LB broth overnight at 37°C with 130-rpm shaking. Genomic DNA was extracted from 500 µL of the culture using the Genomic Micro AX Bacteria Gravity kit (A&A Biotechnology, Gdynia, Poland) according to the manufacturer's instructions. DNA concentration was measured using a Qubit 3.0 fluorometer (Thermo Fisher Scientific). A DNA library was prepared using the Ligation Sequencing Kit SQK-LSK109 with the Native Barcoding Expansion Kits EXP-NBD 103 and EXP-NBD 114 (Oxford Nanopore Technologies, Oxford, UK) according to the instructions, and 1-µg DNA of each strain was sequenced using the MinION MK1B sequencing device (Oxford Nanopore Technologies). FASTQ data were extracted from FAST5 files using the Guppy Basecalling software (v. 3.4.4), and the extracted files were further demultiplexed using Porechop (v. 0.2.4; https://github.com/rrwick/porechop) and filtered by NanoFilt v. 2.7.1 (49) with default parameters.

The *de novo* assembly pipeline Unicycler v.04.9b (29) was used to combine short-read data generated from this and the previous study (18), as well as the long-read sequencing data of each strain generated in this study. This was done to generate complete genome (chromosome and plasmid) sequences. The Unicycler pipeline was employed with the following tools and versions: SPAdes v. 3.13.2 (50), bowtie 2 v. 2.4.1 (51), samtools v. 1.9 (52), racon v. 1.4.16 (https://github.com/lbcb-sci/racon), BLAST (v. 2.9.0+), and java (v. 1.8.0_101). After *de novo* assembly, Unicycler uses Pilon v. 1.23 (53) to polish assembled sequences and exclude sequences shorter than 500 bp.

## Genome annotation and bioinformatic analysis

The genomic DNA of all strains included in this study were bioinformatically circularized using the Unicycler pipeline. Precise species identification of the strains was done by comparing the complete sequences to the type strains of *Citrobacter, Enterobacter, Escherichia,* and *Klebsiella* using the dDDH platform of the Deutsche Sammlung von Mikroorganismen und Zellkulturen using formula 2 (54). Annotation was done using BV-BRC (55) and the NCBI Prokaryotic Genome Annotation Pipeline (v.6.4) (56). All sequences were screened for antibiotic resistance genes, replicon types, and mobile genetic elements using the ResFinder 4.1, PlasmidFinder 2.1, and MobileElementFinder tools available from the Center for Genomic Epidemiology (http://www.genomicepidemiology.org/). ResFinder 4.1 (57–59) was used to find acquired antibiotic resistance genes with default parameters. PlasmidFinder 2.1 was used with the *Enterobacteriaceae* database, using an 80% threshold for minimum identity and a 60% minimum coverage, as previously recommended (59, 60). The MobileElementFinder [software version: v1.0.3 (2020-10-09), database version: v1.0.2 (2020-06-09)] was used to identify insertion sequences and transposons (61). The IntegronFinder (v2.0.2) on the Galaxy Pasteur

platform was used to identify integrons (62, 63). The Mob-suite (version 3.0.0, python 3.9) was used for replicon typing and the prediction of mobility of plasmids (64). Linear comparison of multiple genomic sequences was conducted using Easyfig v. 2.2.2 (65).

## ACKNOWLEDGMENTS

The technical assistance of Jessica Obermeyer, Katja Magnussen, Benedict Staack, Yvonne Wölk, and Adrian Prager is kindly acknowledged.

## AUTHOR AFFILIATION

¹Department of Microbiology and Biotechnology, Max Rubner-Institut, Federal Research Institute for Nutrition and Food, Kiel, Germany

## AUTHOR ORCIDs

Diana Habermann https://orcid.org/0000-0001-9710-5172
Gyu-Sung Cho http://orcid.org/0000-0002-3639-6663

## AUTHOR CONTRIBUTIONS

Maria Stein, Data curation, Investigation, Methodology, Software, Visualization, Writing – original draft, Writing – review and editing | Erik Brinks, Data curation, Software | Jannike Loop, Data curation, Investigation | Diana Habermann, Data curation, Methodology | Charles M. A. P. Franz, Conceptualization, Project administration, Supervision, Writing – original draft, Writing – review and editing.

## DATA AVAILABILITY

Complete genome sequences were deposited in the GenBank/ENA/DDBJ databases under the accession numbers listed in Table S1.

## ADDITIONAL FILES

The following material is available online.

### Supplemental Material

**Table S1 (Spectrum00361-24-s0001.docx).** Complete genome sequences of *Enterobacteriaceae*.

### Open Peer Review

**PEER REVIEW HISTORY (review-history.pdf).** An accounting of the reviewer comments and feedback.

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
