## [Reviewer comments · Microbiology Spectrum]

Microbiology Spectrum

Antibiotic-resistance plasmids in Enterobacteriaceae isolated from fresh produce in northern Germany

Maria Stein, Erik Brinks, Jannike Loop, Diana Habermann, Gyu-Sung Cho, and Charles Franz

Corresponding Author(s): Gyu-Sung Cho, Max Rubner-Institut Bundesforschungsinstitut für Ernährung und Lebensmittel

Review Timeline:

Submission Date:	February 14, 2024
Editorial Decision:	March 21, 2024
Revision Received:	June 20, 2024
Editorial Decision:	July 8, 2024
Revision Received:	July 19, 2024
Accepted:	July 25, 2024

Editor: Adelumola Oladeinde

Reviewer(s): Disclosure of reviewer identity is with reference to reviewer comments included in decision letter(s). The following individuals involved in review of your submission have agreed to reveal their identity: Ryan Blaustein (Reviewer #1)

Transaction Report:

DOI: <https://doi.org/10.1128/spectrum.00361-24>

Re: Spectrum00361-24 (Antibiotic-resistance plasmids in enterobacteria isolated from fresh produce in northern Germany)

Dear Dr. Gyu-Sung Cho:

Thank you for the privilege of reviewing your work. Below you will find my comments, instructions from the Spectrum editorial office, and the reviewer comments.

Revision Guidelines

Sincerely,
Adelumola Oladeinde
Editor
Microbiology Spectrum

Editor:

Please ensure that whole genome sequences analyzed in this study are publicly available on existing databases like NCBI.

Reviewer #1 (Comments for the Author):

see attached

Reviewer #2 (Comments for the Author):

Stein et al described antibiotic resistance plasmids in enterobacteria isolated from fresh produce in northern Germany, determining their antimicrobial susceptibility patterns, and the presence of multidrug resistance from the isolated bacteria. Although the study looks interesting and would add to the existing literature on antimicrobial resistance, it requires substantial revision.

Generally, the manuscript looks good however, some sections have been poorly written. The results and discussion sections require great improvement.

My suggestions are as follows:

1. Preferable to use Enterobacteriaceae as opposed to enterobacteria and be consistent through out the manuscript for ease of reading and clarity.
2. The results section is unnecessarily too long and should be reduced remarkably. The results should be focused on the findings of the present study beginning with the description of the bacteria isolates detected in raw vegetables.
3. All sentences describing what was done previously should be deleted from the manuscript. All sentences in the results section describing what was done in this present study should be moved to the methods section of the manuscript, please.
4. Merge the information in line 193 with line 224 as some of these are mere repetition hence redundant.
5. Sentences describing results that have not been presented in this manuscript should be deleted.
6. In the discussion section, please provide possible explanations for your major findings while comparing these to the findings of other relevant studies.
7. Please highlight some limitations of the present study in the last paragraph of the discussion section.
8. Please provide some recommendations based on the key findings of this study in the last paragraph of the conclusion.

Stein et al. characterized whole genome sequences (WGS) of 22 strains of Enterobacteriaceae isolated from fresh produce and herbs by combining short- and long-read assembly. They recovered plasmids in most genomes (40 total plasmids) and characterized functional potential, including genetic signatures for plasmid transferability, AMR, and heavy metal tolerance. The authors demonstrated links between AMR genotypes and phenotypes among the strains. Overall, the study highlights raw vegetables as a reservoir for diverse Enterobacteriaceae that may carry/transfer genes with implications for public health. It appears to follow their previously published paper that had focused on included short-read sequencing analysis of most of these isolates (coupled with phenotypic assay) (Cho et al. 2021; ref #18 in the text).

General comments:

1. Please clarify rationale for the sequencing effort and strain selection in the introduction. Identifying and characterizing MGEs, especially plasmids, seems to be the main purpose for re-processing isolates with long-read sequencing analysis. Expand commentary on this in the intro.
2. Table 1: Provide isolation dates, which may be relevant to any redundancy in features across isolates. For example, were the Endy1/Endy2 isolates from the same mung bean sample? Same isolation plate? If clonal, why not just use one strain in making comparisons with other Enterobacteriaceae? Similar comments for Cipa6.1/2 and Ciw5.1/2. Also, if these are all clonal, why are there discrepancies (albeit minor) in the AMR-MIC assay shown in Table 2 and mentioned in L. 169-170?
3. Related to #2, when using percentages to explain the frequency of occurrence of genetic features in Enterobacteriaceae, such as plasmids, it may not be appropriate to include clonal strains. For example, 16/19 (84%) of the unique strains appeared to carry plasmids (since Endy1 and Endy2 are the same, and Cip6.1/2 and Ciw5.1/2), though 17/22 (77%) Enterobacteriaceae isolates were reported to carry plasmids (L. 184). Also, were there 40 unique plasmids? Seems like the Ciw5.1/2 have the same properties, including exact bp length. As such, would it be correct to report that there were instead 38 plasmids among the isolates?
4. Please provide info on assembly stats around L. 98 (e.g., total contigs and contig length) and genome content as well (e.g., total CDS in Table 3). How do the new assemblies compare to the previous report? How did you confirm that the coupled approach provided full resolution (L. 327)?
5. Throughout the text, 'metal resistance' should be changed to 'heavy metal resistance' (e.g., L. 283)
6. Please state limitations in the discussion. For example, is comparing 22 genomes and the plasmids they carry reflective of Enterobacteriaceae? These bacteria are prevalent on produce (L. 68) and represent an important and diverse taxonomic group (even the genera and species at the finer scale are highly diverse).
7. Please remember to make your data publicly available with publication. Table S2 accession numbers do not appear in NCBI at the time of this review.

Minor comments:

- L. 58/79: correct the nomenclature; e.g., at first mention of the taxa, remove '(E.)' and '(L.)' and just state *Genus species*.
- L. 107: why not show the results?
- L. 144: Change 'En.' to 'E.'
- L. 203-204: What were examples of the coding sequences on these?
- L. 224: 'only nice strains were detected to have harbored plasmids with resistance genes'
- L. 230: It writes that "there was no relationship between plasmid size and the number of antibiotic resistance gene". What statistical test?
- L. 321: Just state the number. How many of the 22?
- L. 346: Do you mean the same product sample?
- L. 359/405: correct the 'beta' symbols to appear in the same font
- L. 466-468: state why 41.5°C was chosen for incubation
- Please use "bp" or "kbp" consistently; e.g. L. 196 7000 bp, L. 227: 7 kb

**Antibioticresistance plasmids in Enterobacteriaceae isolated from fresh produce in**
**northern Germany**

Maria Stein^a, Erik Brinks^a, Jannike Loop^a, Diana Habermann^a, Gyu-Sung Cho^{a#}, Charles
5 M.A.P. Franz^a

7 ^a Department of Microbiology and Biotechnology, Max Rubner-Institut, Federal Research
Institute for Nutrition and Food, Kiel, Germany

Running Head: Antibiotic-Resistance Plasmids

#Address correspondence to Gyu-Sung Cho, gysung.cho@mri.bund.de

Fax: +49-431-609-2362

Phone: +49-431-609-2340

Email: Gyusung.Cho@mri.bund.de

**KEYWORD:** fresh produce, antibiotic resistance, whole genome sequencing, plasmids,

*Enterobacteriaceae*

ABSTRACT

In this study, the genomes of 22 Enterobacteriaceae isolates from fresh produce and herbs obtained
from retail markets in northern Germany were completely sequenced with MiSeq short read and
MinION long read sequencing and assembled using a Unicycler hybrid assembly. The data showed
that 17 of the strains harbored between one and five plasmids, whereas in five strains only the circular
chromosomal DNA was detected. In total, 40 plasmids were identified. The size of the plasmids
detected varied between ca. 2 and 326 kbp and metal resistance genes were found on seven (17.5%) of
the plasmids. Eleven plasmids (27.5 %) showed the presence of antibiotic resistance genes. Among
large plasmids (> 32 kbp), IncF plasmids (specifically, IncFIB and IncFII) were the most abundant
replicon types, while all small plasmids were Col-replicons. Six plasmids harbored unit and composite
transposons carrying antibiotic resistance genes, with IS26 identified as the primary insertion
sequence. Class 1 integrons carrying antibiotic resistance genes were also detected on chromosomes of
two *Citrobacter* isolates and on four plasmids. Mob-suite analysis revealed that 35% of plasmids in
this study were found to be conjugative, while 30% were identified as mobilizable. Overall, our study
showed that Enterobacteriaceae from fresh produce possess antibiotic resistance genes on both
chromosome and plasmid, some of which are considered to be transferable. This indicates the potential
for Enterobacteriaceae from fresh produce that is usually eaten in raw state to contribute to the transfer
of resistance genes to bacteria of the human gastrointestinal system.

IMPORTANCE

This study showed that Enterobacteriaceae from raw vegetables carried plasmids ranging in size from
2,715 to 326,286 bp, of which about less than one-third carried antibiotic resistance genes encoding
resistance towards antibiotics such as tetracyclines, aminoglycosides, fosfomycins, sulfonamides,
quinolones and beta-lactam antibiotics. Some strains encoded multiple resistances and some encoded
extended spectrum beta-lactamases. The study highlights the potential of produce, that may be eaten
raw, as potential vehicle for transfer of antibiotic resistant bacteria.

**Introduction**

Antibiotic resistance is a global health and developmental threat. As antibiotics are becoming
increasingly ineffective, resistant bacteria can spread globally and lead to infections which are difficult
to treat and may result in death (1, 2). One possible source of antibiotic-resistant pathogens and
opportunistic pathogens is food, in particular animal meats (3). In contrast to food products originating
from animals, fresh produce has also come into focus as a reservoir for antibiotic resistant and/or
pathogenic bacteria (4-7). Vegetables, fruits, fresh cut products and sprouts are considered part of a
healthy diet because they supply a combination of vitamins, antioxidants and minerals (8). However,
because of the heat instability of the nutritional compounds, produce is often consumed minimally
processed or raw and contaminating bacteria are therefore not inactivated (9). Fresh produce can
therefore cause widespread disease outbreaks when these are contaminated with pathogens along the
farm-to-fork-route (8). The most common human pathogenic bacteria associated with fresh produce
are *Escherichia (E.) coli*, *Listeria (L.) monocytogenes* and *Salmonella* (8, 10). For example,
*Salmonella* Coeln caused an outbreak associated with ready-to-eat salads in Norway (11), fresh
bagged spinach contaminated by *E. coli* O157:H7 led to a multistate outbreak in the USA in 2006 (12)
and packaged leafy greens contaminated with *L. monocytogenes* caused a listeriosis outbreak in USA
and Canada (13). In Germany, unprocessed fresh produce was brought more into public focus as a
potential vehicle for pathogenic bacteria after the *E. coli* O104:H4 outbreak caused by contaminated
sprouts in northern Germany in 2011, where 54 people died (14, 15).

Fiedler et al. (16) examined 200 fresh produce samples from markets in northern Germany and
reported a low incidence of pathogenic bacteria in these products. Despite this, the mean aerobic
mesophilic bacterial counts were considerably high (7 to 8 log₁₀ cfu/g) and the *Enterobacteriaceae*
counts varied greatly within the sampled products and ranged from 2 to 7.5 log₁₀ cfu/g in leaf lettuce
(16). This indicates that fresh produce may be a potential vehicle for opportunistic pathogens,
including potentially antibiotic-resistant enterobacteria that may contribute to the spread of antibiotic
resistance genes via this food route (17, 18). In a previous study, Blau et al. (19) isolated tetracycline-
resistant *E. coli* from mixed salads, cilantro and arugula from the German market and showed that
these carried IncF, Inc11, IncN, IncHI1, IncU, IncP-1 β and IncX1 plasmids. Furthermore, conjugative

plasmids encoding tetracycline resistance were captured by exogenous plasmid isolation using an *E.*
*coli* recipient strain (19). The study thus emphasized the role of conjugative plasmids in enterobacteria
in horizontal gene transfer that might take place on fresh produce, adding to the spread of antibiotic
resistance genes.

Our previous study showed that potentially opportunistic pathogens, belonging mostly to *E.*
*coli*, *Klebsiella* (*K.*) *pneumoniae*, *Citrobacter* (*C.*) *portucalensis* and *Enterobacter* (*En.*) *ludwigii*,
could be isolated from fresh produce in Germany, and that amongst these, strains occurred which were
resistant to multiple antibiotic classes (18). In this study, we aimed to further characterize 22 strains
that were isolated from fresh produce by genome sequencing, which combined short read (MiSeq) and
long read (MinION) sequencing data in a hybrid assembly to generate complete chromosome and
plasmid sequences. This investigation thus aimed to gain a better understanding of the diversity of
antibiotic resistance genes and their genetic location in strains of fresh produce origin. Furthermore,
we aimed to investigate the diversity of the plasmids present in these strains and the potential role of
the plasmids for spreading antibiotic resistance genes across the farm-to-fork-route.

**Results**

The 22 *Enterobacteriaceae* strains of this study thus consisted of *Citrobacter* spp. (n=8),
*Enterobacter* spp. (n=4), *Escherichia coli* (n=4) and *Klebsiella* spp. (n=6) (Table 1). All of these
genera are known to contain species of importance as opportunistic nosocomial pathogens, and are
often antibiotic resistant (21). Despite showing 100 % identity when compared using dDDH and
having identical plasmids, Ciw5.1 showed an approximately 8809-bp larger chromosome than Ciw5.2.
Notably, a large antibiotic resistant gene region was present in strain Ciw5.1 but not in Ciw5.2, and
this region and surrounding sequences are indicated using Easyfig (v.2.2.2) (Fig. 1).

**Phenotypic antibiotic resistance.** The phenotypic antibiotic resistances are shown in Table 2.
Nineteen out of 22 strains (86 %) showed resistance to ampicillin, while two strains (9 %) showed an
intermediate resistance. Thus, only one strain, i.e. *E. coli* Ec1120, was susceptible to ampicillin. The
second most commonly-occurring resistance was against tetracycline, as 12 of the 22 strains (55 %)

were resistant. Eight strains (36 %) showed resistance to streptomycin and seven strains (32 %) to
ciprofloxacin (Table 2). Four strains (18 %) showed resistance to chloramphenicol and to cefotaxime,
and 2 strains to gentamicin (9 %). Eleven of the 22 strains (50 %) showed multiple resistance to at
least 3 different classes of antibiotics (mainly against tetracyclines, aminoglycosides and beta
lactams), while 3 strains showed resistance to at least six antibiotics (tetracycline, streptomycin,
chloramphenicol, ampicillin, ciprofloxacin and cefotaxime). The *K. pneumoniae* strain Kpneu34 and
*C. werkmanii* strain Ciw5.1, both isolated from different sprout samples, were the only two strains that
showed resistance towards gentamicin and also to all of the antibiotics tested (Table 2).

**Genomic characterization.** The largest contig obtained from each strain genomic sequencing
represented the complete bacterial chromosomal DNA (> 4.5 Mbp in all cases). The complete DNA
sequences of 22 enterobacterial strains were used to identify the bacteria to species level by dDDH
(Table 1) and the *Citrobacter* strains were found to include the species *C. gillenii* (n=1), *C.*
*portucalensis* (n=2), *C. werkmanii* (n=2), *C. pasteurii* (n=2) and *C. freundii* (n=1), the *Enterobacter*
strains included the species *En. dykesii* (n=2), *En. hormaechei* (n=1) and *En. bugandensis* (n=1), the
*Klebsiella* strains included the species *K. grimontii* (n=1), *K. variicola* (n=1) and *K. pneumoniae* (n=4)
while the *Escherichia* strains included only the species *E. coli* (n=4).

The genome sizes of the strains in this study varied from 4.55 Mbp to 5.85 Mbp, with *K.*
*grimontii* exhibiting the largest and the two *En. dykesii* strains having the smallest genomes of all
investigated strains (Table 3). When considering the *Citrobacter* spp., the chromosomal DNA sizes
ranged from 4.69 Mbp to 5.09 Mbp, with the two *C. pasteurii* strains having the smallest genomes of
4.69 Mbp. The *Enterobacter* spp. genome sizes ranged from 4.55 Mbp (*En. dykesii*) to 4.84 Mbp (*En.*
*bugandensis*), while those for *E. coli* strains ranged from 4.58 Mbp to 4.93 Mbp. Overall, the *Klebsiella*
genomes were the largest chromosomes in this study ranging from 5.26 Mbp (*K. pneumoniae*) to 5.85
Mbp (*K. grimontii*) (Table 3). The chromosomal mol% GC contents ranged from 50.79 to 57.49 %
(differing by a maximum of 6.7 mol%), with *E. coli* Ec1119 having the lowest and *K. pneumoniae*
Kpneu28 having the highest mol% GC content. The mol% GC content varied noticeably between
genera, but were similar within a genus. *Escherichia* and *Citrobacter* had rather low mol% GC

contents ranging from 50.79 to 50.88 % and 51.63 to 52.5 %, respectively, while *Enterobacter* and
*Klebsiella* showed higher mol% GC contents ranging from 55.7 to 56.07 % and 55.89 to 57.49 %,
respectively.

For each of the three pairs of strains showing high similarity in chromosomal DNA size
(Endy1 and Endy2, Cipa6.1 and Cipa6.2, as well as Ciw5.1 and Ciw5.2), the dDDH analysis resulted
in 100 % identity, despite length differences of 3 bp, 1471 bp and 8809 bp, respectively. The strains
Endy1 and Endy2, as well as Cipa6.1 and Cipa6.2 showed similar phenotypic antibiotic resistance
patterns, were isolated from the same fresh produce (Table 1 and 2), and were identified as clonal
isolates. Strains Ciw5.1 and Ciw5.2 were also isolated from the same product, i.e. China rose sprouts
(Table 1). The nucleotide differences between Ciw5.1 (5.08 Mbp) and Ciw5.2 (5.07 Mbp) occurred at
different chromosomal locations, of which the largest coherent differing sequence of 8152 bp was
inspected more closely (Fig. 1). The data showed that there was an insertion/deletion of a resistance
region, including an integron. Strain Ciw5.2 possessed one class 1 integrase, while strain Ciw5.1
possessed two class 1 integrases and nine additional genes in between those two integrases. These
encoded two hypothetical proteins, two IS91 insertion sequences, a disinfectant resistance gene, and
four antibiotic resistance genes (*dfra19*, *catB3*, *sul1*, and *ant(2'')-Ia*) (Fig. 1). At the phenotypic level,
the absence of *ant(2'')-Ia* in Ciw5.2 led to the strain being sensitive towards gentamicin, while Ciw5.1
showed resistance to this antibiotic. The chloramphenicol MIC value for both strains was very high
(>256 µg/ml), because of *catA1* gene, which was present in both strains (Table 2).

Five of the 22 strains (Cipa6.1, Cipa6.2, Endy1, Endy2, and Kva3) did not harbor any
extrachromosomal DNA sequences. The remaining 17 strains (77 %) harbored in total 41
extrachromosomal DNA sequences, ranging from ca. 2 kbp to ca. 326 kbp (Table 3). One of the
extrachromosomal DNA sequences of the strain Enb12 was identified as a self-replicating prophage.
Most of this 45,550 bp sequence was classified as an intact prophage (score=105) with 58 protein
coding sequences. This Enb12 DNA sequence showed the highest similarity to prophage N15 in the
blastn analysis. This prophage N15 was characterized as an extrachromosomal, linear prophage (25,
26). The Enb12 prophage was named MStein-2023a upon submission of the sequence to the NCBI
GenBank database and was excluded from further description of plasmids in this study (Table 3).

The 40 plasmids detected in the 17 strains occurred at numbers of between one and five
 different plasmids per strain. Mostly strains harbored one (6 strains) or two (5 strains) plasmids (Table
 3). In this study, approx. one third (14 plasmids, 35 %) of plasmids were considered small plasmids
 (defined here as < 7,000 bp), while the remaining approx. two-thirds (26 plasmids, 65 %) were
 considered large plasmids (defined here as > 32,000 bp). Except for strains Ciw5.1 and Ciw5.2, all
 *Citrobacter* plasmids were rather large plasmids (> 32,000 bp). Cipo13 was the only strain harboring
 five plasmids, which were all considered large, amounting to a total of 0.6 Mbp of extrachromosomal
 DNA size (Table 3). All of the *Enterobacter* and *Escherichia* strains, except for Ec1115, also
 possessed predominantly large plasmids. Strain Ec1115 had one small as well as three large plasmids.
 The *K. pneumoniae* strains had a variety of plasmid sizes, including generally one large and one small
 plasmid (Table 3). In total, 27.5 % (11/40) of the plasmids carried antibiotic resistance genes, whereby
 it was noted that none of the small plasmids carried such genes.

All *Citrobacter* strains, except for the strains Cig11, Cipa6.1 and Cipa6.2 possessed resistance
 genes on their chromosome. The AmpC β -lactamase gene *bla_{CMY}* was found in five *Citrobacter*
 chromosomes, while the quinolone resistance gene *qnrB* was found in four out of eight *Citrobacter*
 strains. With 10 antibiotic resistance genes (*tet(B)*, *bla_{CMY-98}*, *bla_{OXA-1}*, *sul1*, *qnrB34*, *catB3*, *catA1*,
 *dfrA19*, *aadA1*, and *ant(2'')-Ia*), potentially conferring resistance to seven antibiotic resistance classes
 (tetracycline, β -lactam, sulfonamides, diaminopyrimidines, quinolone, phenicol and aminoglycoside
 antibiotics), *C. werkmanii* strain Ciw5.1 possessed the most diverse antibiotic resistance genes present
 on chromosomal DNA in this study (Fig. 1). Antibiotic resistance genes on *Enterobacter*
 chromosomes were similar, because all possessed the *fosA* gene while three of the four *Enterobacter*
 chromosomes also possessed the *bla_{ACT-6}* or *bla_{ACT-7}* genes (Table 3). On all four *E. coli* genomes, no
 resistance genes could be detected using ResFinder. All *Klebsiella* strains except strain Kgr1 encoded
 *fosA*, *oqxA* and *oqxB* in combination with one of the β -lactam antibiotic resistance genes, i.e. *bla_{SHV}*
 or *bla_{LEN}* on their chromosome. Generally, *Enterobacter* spp. and *Klebsiella* spp. resembled each other
 the most regarding the antibiotic resistance genes that they carried on the chromosome. These genera
 also showed general phenotypic resistance to ampicillin, which could possibly be explained by the

[revised manuscript text omitted]

The copy numbers of specific, potentially conjugative plasmids carrying antibiotic-resistance
genes present in each of the strains Kpneu28, Kpneu34, Ec1115 and Ec1117 was determined using
qPCR. The copy numbers of the four IncF plasmids were between 1.8 and 3.8 copies per chromosome
(Table 5).

The two plasmids of the *Escherichia* strains exhibited lower copy numbers when compared to
the *Klebsiella* plasmids (Table 5). The smallest tested plasmid (pEC1117_1) (119,797 bp) had the
lowest determined copy number of 1.8 / chromosome, while the plasmid with the highest copy number
(3.8 copies / chromosome; pKPNEU28_2) was only slightly larger (121,449 bp) than pEC1117_1.

Discussion

[revised manuscript text omitted]

**Determination of plasmid copy number.** A qPCR method was used to calculate the copy number for
plasmids pEC1115_1, pEC1117_1, pKPNEU28_2 and pKPNEU34_1. These plasmids were selected
for copy number determination as they showed an IncF replicon type (PlasmidFinder) combined with
predicted conjugative properties and mating pair formation type mpfF (*mob*-suite), indicating that they
were potentially transferable. Furthermore, they encoded antibiotic resistance genes, as identified by
ResFinder. Therefore, these strains could serve as potential candidates for conjugation experiments in
future studies. For copy number determination, qPCR primers were designed for each plasmid from
the plasmid sequence determined in this study (Suppl. Table 1)

The primers were designed to target gene sequences encoding part of the replication protein
(IncFIB replication protein (pEC1115_1, pEC1117_1, and pKPNEU34_1) and IncFII replication
protein (pKPNEU28_2). Primers were also designed to amplify the beta-subunit of the RNA
polymerase (*rpoB*) housekeeping gene (Suppl. Table 2) as a chromosomal reference gene. Primers
were tested in a conventional PCR approach and the product sizes on the gel were compared to the
sizes of the targeted fragments. For qPCR, three separate 20 ml overnight cultures were prepared for
each strain of Ec1115, Ec1117, Kpneu28 and Kpneu34 to obtain three biological replicates (incubating
at 37°C with shaking at 130 rpm, LB-broth). The culture was diluted ten-fold with PCR grade water

(Roth) and the dilution was subsequently heated at 98°C for 10 minutes followed by an incubation step
at -20 °C for 20 minutes (51). Supernatant was diluted tenfold to dilute PCR inhibitors, as this was
found to improve the amplification by qPCR. qPCR was performed in a 10 µl volume, and the reaction
mixture contained 200 nM of each primer (Table 2), 5 µl iTaq Universal SYBR Green Supermix (Bio-
Rad, Munich, Germany), 2 µl of heat inactivated DNA from tenfold diluted supernatant and 2.6 µl
PCR grade H₂O. The qPCR was performed in a technical triplicate, to get 9 measurements in total for
each strain. The qPCR was performed using a 2-step program in a CFX96 Touch Real-Time PCR
Detection System (Bio-Rad), with an initial heating at 94°C for 3 minutes followed by 40 cycles of 10
530 s at 94°C and 1 min at 60°C and included a melting curve analysis (65°C – 95°C, 0.5°C per cycle). For
each strain, *rpoB* and plasmid detection was performed simultaneously, in reactions stemming from
the same overnight culture. In order to calculate amplification efficiency, each primer set was used for
generating standard curves with tenfold serially diluted DNA as template. The copy number of each
plasmid (PCN) was calculated relative to chromosome copy number using the following equation
(51): $E = 10^{(-1/\text{slope})}$ and $\text{PCN} = (\text{Eca})^{\text{Ctc}}/(\text{Epa})^{\text{Ctp}}$, where Eca = chromosome amplification efficiency
and Epa = plasmid amplification efficiency.

**Data availability**

[revised manuscript text omitted]

- 49. Smillie C, Garcillán-Barcia MP, Francia MV, Rocha EP, de la Cruz F. 2010. Mobility of plasmids.
*Microbiol Mol Biol Rev* 74:434-52.10.1128/mmbr.00020-10
- 50. Ogier J-C, Pagès S, Galan M, Barret M, Gaudriault S. 2019. *rpoB*, a promising marker for
analyzing the diversity of bacterial communities by amplicon sequencing. *BMC Microbiology*
19:171.10.1186/s12866-019-1546-z
- 51. Skulj M, Okrslar V, Jalen S, Jevsevar S, Slanc P, Strukelj B, Menart V. 2008. Improved
determination of plasmid copy number using quantitative real-time PCR for monitoring
fermentation processes. *Microb Cell Fact* 7:6.10.1186/1475-2859-7-6
- 52. FDA. 2021. NARMS Interpretive Criteria for Susceptibility Testing (accessed May 3, 2022)
(<https://www.fda.gov/media/108180/download>).
- 53. Bolger AM, Lohse M, Usadel B. 2014. Trimmomatic: a flexible trimmer for Illumina sequence
data. *Bioinformatics* 30:2114-20.10.1093/bioinformatics/btu170
- 54. De Coster W, D'Hert S, Schultz DT, Cruts M, Van Broeckhoven C. 2018. NanoPack: visualizing
and processing long-read sequencing data. *Bioinformatics* 34:2666-
2669.10.1093/bioinformatics/bty149
- 55. Bankevich A, Nurk S, Antipov D, Gurevich AA, Dvorkin M, Kulikov AS, Lesin VM, Nikolenko SI,
Pham S, Prjibelski AD, Pyshkin AV, Sirotkin AV, Vyahhi N, Tesler G, Alekseyev MA, Pevzner PA.
2012. SPAdes: a new genome assembly algorithm and its applications to single-cell
sequencing. *J Comput Biol* 19:455-77.10.1089/cmb.2012.0021
- 56. Langmead B, Salzberg SL. 2012. Fast gapped-read alignment with Bowtie 2. *Nat Methods*
9:357-9.10.1038/nmeth.1923
- 57. Danecek P, Bonfield JK, Liddle J, Marshall J, Ohan V, Pollard MO, Whitwham A, Keane T,
McCarthy SA, Davies RM, Li H. 2021. Twelve years of SAMtools and BCFtools. *Gigascience*
10.10.1093/gigascience/giab008
- 58. Walker BJ, Abeel T, Shea T, Priest M, Abouelliel A, Sakthikumar S, Cuomo CA, Zeng Q,
Wortman J, Young SK, Earl AM. 2014. Pilon: an integrated tool for comprehensive microbial
variant detection and genome assembly improvement. *PLoS One*
9:e112963.10.1371/journal.pone.0112963
- 59. Meier-Kolthoff JP, Auch AF, Klenk HP, Goker M. 2013. Genome sequence-based species
delimitation with confidence intervals and improved distance functions. *BMC Bioinformatics*
14:60.10.1186/1471-2105-14-60
- 60. Olson RD, Assaf R, Brettin T, Conrad N, Cucinell C, Davis JJ, Dempsey DM, Dickerman A,
Dietrich EM, Kenyon RW, Kuscuoglu M, Lefkowitz EJ, Lu J, Machi D, Macken C, Mao C,
Niewiadowska A, Nguyen M, Olsen GJ, Overbeek JC, Parrello B, Parrello V, Porter JS, Pusch
GD, Shukla M, Singh I, Stewart L, Tan G, Thomas C, VanOeffelen M, Vonstein V, Wallace ZS,
Warren AS, Wattam AR, Xia F, Yoo H, Zhang Y, Zmasek CM, Scheuermann RH, Stevens RL.
2023. Introducing the Bacterial and Viral Bioinformatics Resource Center (BV-BRC): a
resource combining PATRIC, IRD and ViPR. *Nucleic Acids Res* 51:D678-
d689.10.1093/nar/gkac1003
- 61. Li W, O'Neill KR, Haft DH, DiCuccio M, Chetvernin V, Badretdin A, Coulouris G, Chitsaz F,
Derbyshire MK, Durkin AS, Gonzales NR, Gwadz M, Lanczycki CJ, Song JS, Thanki N, Wang J,
Yamashita RA, Yang M, Zheng C, Marchler-Bauer A, Thibaud-Nissen F. 2021. RefSeq:
expanding the Prokaryotic Genome Annotation Pipeline reach with protein family model
curation. *Nucleic Acids Res* 49:D1020-D1028.10.1093/nar/gkaa1105
- 62. Bortolaia V, Kaas RS, Ruppe E, Roberts MC, Schwarz S, Cattoir V, Philippon A, Allesoe RL,
Rebelo AR, Florensa AF, Fagelhauer L, Chakraborty T, Neumann B, Werner G, Bender JK,
Stingl K, Nguyen M, Coppens J, Xavier BB, Malhotra-Kumar S, Westh H, Pinholt M, Anjum MF,
Duggett NA, Kempf I, Nykasenoja S, Olkkola S, Wiczorek K, Amaro A, Clemente L, Mossong J,
Losch S, Ragimbeau C, Lund O, Aarestrup FM. 2020. ResFinder 4.0 for predictions of
phenotypes from genotypes. *J Antimicrob Chemother* 75:3491-3500.10.1093/jac/dkaa345
- 63. Zankari E, Allesoe R, Joensen KG, Cavaco LM, Lund O, Aarestrup FM. 2017. PointFinder: a
novel web tool for WGS-based detection of antimicrobial resistance associated with

- chromosomal point mutations in bacterial pathogens. *J Antimicrob Chemother* 72:2764-
2768.10.1093/jac/dkx217
- 64. Camacho C, Coulouris G, Avagyan V, Ma N, Papadopoulos J, Bealer K, Madden TL. 2009.
BLAST+: architecture and applications. *BMC Bioinformatics* 10:421.10.1186/1471-2105-10-
421
- 65. Carattoli A, Zankari E, Garcia-Fernandez A, Voldby Larsen M, Lund O, Villa L, Moller Aarestrup
F, Hasman H. 2014. In silico detection and typing of plasmids using PlasmidFinder and
plasmid multilocus sequence typing. *Antimicrob Agents Chemother* 58:3895-
903.10.1128/AAC.02412-14
- 66. Johansson MHK, Bortolaia V, Tansirichaiya S, Aarestrup FM, Roberts AP, Petersen TN. 2021.
Detection of mobile genetic elements associated with antibiotic resistance in *Salmonella*
*enterica* using a newly developed web tool: MobileElementFinder. *J Antimicrob Chemother*
76:101-109.10.1093/jac/dkaa390
- 67. Neron B, Littner E, Haudiquet M, Perrin A, Cury J, Rocha EPC. 2022. IntegronFinder 2.0:
Identification and Analysis of Integrons across Bacteria, with a Focus on Antibiotic Resistance
in *Klebsiella*. *Microorganisms* 10.10.3390/microorganisms10040700
- 68. Cury J, Jove T, Touchon M, Neron B, Rocha EP. 2016. Identification and analysis of integrons
and cassette arrays in bacterial genomes. *Nucleic Acids Res* 44:4539-50.10.1093/nar/gkw319
- 69. Robertson J, Nash JHE. 2018. MOB-suite: software tools for clustering, reconstruction and
typing of plasmids from draft assemblies. *Microb Genom* 4.10.1099/mgen.0.000206
- 70. Sullivan MJ, Petty NK, Beatson SA. 2011. Easyfig: a genome comparison visualizer.
*Bioinformatics* 27:1009-10.10.1093/bioinformatics/btr039

Stein et al. characterized whole genome sequences (WGS) of 22 strains of Enterobacteriaceae isolated from fresh produce and herbs by combining short- and long-read assembly. They recovered plasmids in most genomes (40 total plasmids) and characterized functional potential, including genetic signatures for plasmid transferability, AMR, and heavy metal tolerance. The authors demonstrated links between AMR genotypes and phenotypes among the strains. Overall, the study highlights raw vegetables as a reservoir for diverse Enterobacteriaceae that may carry/transfer genes with implications for public health. It appears to follow their previously published paper that had focused on included short-read sequencing analysis of most of these isolates (coupled with phenotypic assay) (Cho et al. 2021; ref #18 in the text).

General comments:

1. Please clarify rationale for the sequencing effort and strain selection in the introduction. Identifying and characterizing MGEs, especially plasmids, seems to be the main purpose for re-processing isolates with long-read sequencing analysis. Expand commentary on this in the intro.

Thank you very much for pointing this out.
Now this paragraph was rephrased.

Our previous study showed that the presence of potentially opportunistic pathogens, belonging mostly to E. coli, Klebsiella (K.) pneumoniae, Citrobacter (C.) portucalensis and Enterobacter (En.) ludwigii, isolated from fresh produce in Germany. Amongst these, strains resistant to multiple antibiotic classes were identified (18). In this study, we aimed to obtain complete chromosome and plasmid sequences from 22 strains isolated from fresh produce using a hybrid assembly approach, which combined short read (MiSeq) and long read (MinION) sequencing data. Additionally, we utilize various database including ResFinder, PlasmidFinder, and Mob-suite to characterize mobile genetic elements (MGEs), particularly plasmids, within the complete genome sequences. This investigation thus aimed to gain a better understanding of the diversity of antibiotic resistance genes and their genetic location in strains of fresh produce origin. Furthermore, we aimed to investigate the diversity of the plasmids present in these strains and the potential role of the plasmids for spreading antibiotic resistance genes across the farm-to-fork-route.

2. Table 1: Provide isolation dates, which may be relevant to any redundancy in features across isolates.

For example, were the Endy1/Endy2 isolates from the same mung bean sample? Same isolation plate? If clonal, why not just use one strain in making comparisons with other Enterobacteriaceae? Similar comments for Cipa6.1/2 and Ciw5.1/2. Also, if these are all clonal, why are there discrepancies (albeit minor) in the AMR-MIC assay shown in Table 2 and mentioned in L. 169-170?

Thank you for bringing this to our attention.

Yes, you're correct. Each strain, Endy1 and Endy2, Cipa6.1 and 2, and Ciw5.1 and 2 have been isolated from the same samples and same agar plate. However, in the previous study (Cho et al., 2021), there were slight differences in the physiological features of each strain and the Sanger sequencing of 16S rRNA gene sequence of Endy1 were identified as *Enterobacter cancerogenus* with 99.5% sequence similarity while the Endy2 was identified as *Enterobacter asburiae* with 99.65% sequence similarity. That is why in that previous study (18) the strains were still considered different strains. However, when we additionally conducted *in silico* strain identification of these strains in this study using complete genome sequence data, we found that we could not distinguish between them

Therefore, we assigned these to represent clonal isolates. Nevertheless, each of the three pairs of strains (Endy1 and Endy2, Cipa6.1 and Cipa6.2, as well as Ciw5.1 and Ciw5.2) showed slight difference in chromosomal DNA size of 3 bp, 1471 bp and 8809 bp, respectively, this is why we would like to keep them in the Table 3. However, as the plasmids were identical for Ciw5.1 and Ciw5.2 (see comment below) we thankfully take up the suggestion by the reviewer and report the two plasmids of one strain of the clonally related pairs instead of each 2 plasmids of both of the strains, as these showed no difference and the strains were clearly highly related.

As mentioned above, In this study we did however identify differences in antibiotic resistance features between Ciw5.1 and Ciw5.2 due to the insertion/ deletion of integron integrase and other genetic elements, including transposases and antibiotic resistance genes (line 178 to 185 and 340 to 349).

3. Related to #2, when using percentages to explain the frequency of occurrence of genetic features in Enterobacteriaceae, such as plasmids, it may not be appropriate to include clonal strains. For example, 16/19 (84%) of the unique strains appeared to carry plasmids (since Endy1 and Endy2 are the same, and Cip6.1/2 and Ciw5.1/2), though 17/22 (77%) Enterobacteriaceae isolates were reported to carry plasmids (L. 184). Also, were there 40 unique plasmids? Seems like the Ciw5.1/2 have the same properties, including exact bp length. As such, would it be correct to report that there were instead 38 plasmids among the isolates?

Thank you for pointing this out, this is a very valid point. Cip6.1 and Cip6.2 and Endy1 and Endy2 strains don't possess any plasmid related sequences, whereas Ciw5.1 and 2 contain two plasmid related sequences each. Consequently, the number of unique plasmid sequences in this study was not 40 but 38, as these were now reported for only one of the strains. In light of this finding, all percentage values were recalculated and updated in the manuscript.

4. Please provide info on assembly stats around L. 98 (e.g., total contigs and contig length) and genome content as well (e.g., total CDS in Table 3). How do the new assemblies compare to the previous report? How did you confirm that the coupled approach provided full resolution (L. 327)?

Thank you for pointing this.
Now the text is changed as follows.

The results in Table 1 confirm the findings of the previous study, where MiSeq data were used to identify strains using various genotyping and whole genome analysis approaches, excluding strains Cif11, Kva3, Kpneu8, Kpneu28 and Kpneu34, which were newly determined in this study (18). Additional information obtained from complete genome sequence data, such as total CDSs, number of contigs, contig length, and mol% GC contents, is presented in table 2.

Generally speaking, the sequence data were very similar. Small differences did occur as the combined use of MiSeq and Nanopore sequence data gives a more accurate assembly and hence more accurate genome statistics data. GenBank (NCBI) always uses the latest data linked to one submission, thus the old MiSeq sequence data were automatically replaced by the complete genome sequence data linked to this entry by GenBank. The CDS now indicated are the total CDS as calculated by Genbank which relate to both the chromosomal DNA as well as plasmid DNA CDS.

The newly assembled sequences were identified as circular and gap-free through *in silico* analysis using the Pilon pipeline integrated with the Unicycler pipeline. The Pilon pipeline together with highly accurate short reads, is able to close any remaining gaps, indels, or mismatches after de novo

assembly. This indicated complete resolution. In this study, after polishing using Pilon, we found no gaps or mismatches. Therefore, we considered our sequence contigs to be complete. I hope that this explanation sufficiently describes how we confirmed the complete genome sequences.

5. Throughout the text, 'metal resistance' should be changed to 'heavy metal resistance' (e.g., L. 283)

"Metal resistance" was changed to "heavy metal resistance"

6. Please state limitations in the discussion. For example, is comparing 22 genomes and the plasmids they carry reflective of Enterobacteriaceae? These bacteria are prevalent on produce (L. 68) and represent an important and diverse taxonomic group (even the genera and species at the finer scale are highly diverse).

We added the following paragraph:

In this study, we focused on 12 species belonged to Enterobacteriaceae from fresh produce in northern Germany and generated only 22 complete genome sequences. However, the microbiota of fresh produce, including agricultural soil and irrigation water, are much more complex and also highly diverse. Furthermore, while we conducted complete genome sequences, we did not enumerate antibiotic-resistant bacteria present on the fresh produce. Therefore, the 12 species considered in this study may not provide sufficient data on the potential of antibiotic resistance plasmid spread from fresh produce and more extensive research on this would clearly be required.

7. Please remember to make your data publicly available with publication. Table S2 accession numbers do not appear in NCBI at the time of this review.

Thank you for bringing this to our attention.

All nucleotide accession numbers have been updated, and the nucleotide information is now fully available.

Minor comments:

- L. 58/79: correct the nomenclature; e.g., at first mention of the taxa, remove '(E.)' and '(L.)' and just state *Genus species*.

Done as suggested

- L. 107: why not show the results?

Actually, we indicated the dDDH values as 100% in this sentence. The phrase "data not shown" indicate that we did not prepare any further tables or figures for dDDH comparisons among our own isolates. Therefore, we removed the phrase "data not shown" from this sentence.

- L. 144: Change 'En.' to 'E.'

In this study, we have abbreviated *Escherichia* to E. and *Enterobacter* to En. Therefore, in line 144 this was indicated to define *Enterobacter* strains not *Escherichia* strains. Therefore, we prefer to use En. in this context.

- L. 203-204: What were examples of the coding sequences on these?

Thank you very much for pointing this out. Two examples of CDS were added, and the sentence was rephrased.

In total, 27.5 % (11/40) of the plasmids carried antibiotic resistance genes (i.e. tet(A) and sul2). It is noted that none of the small plasmids carried such genes.

- L. 224: 'only nice strains were detected to have harbored plasmids with resistance genes'

Yes, you are right. The strains Cigi1, Cipo4, Cipo13, Ec1115, Ec1117, Ec1120, Kpneu8, Kpneu28, and Kpneu34 harbored plasmids sequences containing antibiotic resistance gene(s) (Table 3). The remaining 8 strains possessed antibiotic resistance gene(s) on their chromosomal DNA.

- L. 230: It writes that "there was no relationship between plasmid size and the number of antibiotic resistance gene". What statistical test?

No this was not statistically tested, we rephrased the wording to...there was no noticeable relationship....indicating that the statement was based on by observation and not on statistical testing.

- L. 321: Just state the number. How many of the 22?

The sentence was changed and rephrased.

Eighteen out of 22 strains in this study were previously sequenced using MiSeq (18).

- L. 346: Do you mean the same product sample?

Yes, you are right. These two strains were isolated from same product sample.

- L. 359/405: correct the 'beta' symbols to appear in the same font

It changed.

- L. 466-468: state why 41.5°C was chosen for incubation

This enrichment temperature selected our study was optimized to detect human pathogenic *E. coli* and *Salmonella* strains in the previous studies (Fiedler et al., 2017 Presence of Human Pathogens in Produce from Retail Markets in Northern Germany; Lamparter et al., 2020; Using hydrochloric acid and bile resistance for optimized detection and isolation of Shiga toxin-producing *Escherichia coli* (STEC) from sprouts; Hara-Kudo et al., 2000; *Escherichia coli* O26 detection from foods using an enrichment procedure and an immunomagnetic separation method).

For clarity and readability, the sentence was changed and rephrased.

*For specific enrichment pathogenic *E. coli* and *Salmonella*, the sample was incubated at 41.5°C for 24 h and again homogenized in a stomacher as done before. One ml of the sample was diluted 1:10 in Brilliant-Green Bile Lactose broth (BRILA) (Merck) and incubated for an additional 24 h at 41.5°C. The increase in temperature serves to reduce the accompanying bacterial microbiota. After incubation, 10 µl of the enriched sample was spread onto Brilliance ESBL agar to detect ESBL-producing*

Enterobacteriaceae, and incubated at 41.5°C for 24 h. Green colonies indicative of extended spectrum β -lactamase production on Brilliance ESBL agar plate were selected and purified by repeated streaking onto the same medium.

- Please use “bp” or “kbp” consistently; e.g. L. 196 7000 bp, L. 227: 7 kb

They are changed to “bp”.

Reviewer #2 (Comments for the Author):

Stein et al described antibiotic resistance plasmids in enterobacteria isolated from fresh produce in northern Germany, determining their antimicrobial susceptibility patterns, and the presence of multidrug resistance from the isolated bacteria. Although the study looks interesting and would add to the existing literature on antimicrobial resistance, it requires substantial revision.

Generally, the manuscript looks good however, some sections have been poorly written. The results and discussion sections require great improvement.

My suggestions are as follows:

1. Preferable to use Enterobacteriaceae as opposed to enterobacteria and be consistent through out the manuscript for ease of reading and clarity.

All enterobacteria were changed to Enterobacteriaceae.

2. The results section is unnecessarily too long and should be reduced remarkably. The results should be focused on the findings of the present study beginning with the description of the bacteria isolates detected in raw vegetables.

Thank you very much for pointing this out.

The results section was reduced and rephrased.

3. All sentences describing what was done previously should be deleted from the manuscript. All sentences in the results section describing what was done in this present study should be moved to the methods section of the manuscript, please.

The manuscript was reduced and rephrased.

4. Merge the information in line 193 with line 224 as some of these are mere repetition hence redundant.

Now this paragraph was merged and shortened.

Most Citrobacter strains, except for the strains Cigi1, Cipa6.1 and Cipa6.2 possessed resistance genes on their chromosome. Notably, C. werkmanii strain Ciw5.1 possessed the most diverse antibiotic resistance genes on chromosomal DNA (10 genes conferring resistance to 7 antibiotic classes) (Fig. 1 and Table 3). Enterobacter spp. and Klebsiella spp. showed similar resistance gene patterns, with fosA and β -lactam resistance genes predominating on their chromosome, while E. coli strains did not exhibit resistance genes on their chromosome. Overall, the most prevalent antibiotic resistance genes on chromosomes of the 22 Enterobacteriaceae strains were quinolone, β -lactam and fosfomycin resistance genes in this study (Table 2 and 3).

5. Sentences describing results that have not been presented in this manuscript should be deleted.

It has now been changed.

The text from line 160-172 has been deleted.

In the previous study, most of the strains were identified with draft genome sequences obtained from MiSeq sequencing. The complete chromosomal DNA sequences of the same strains were obtained in this study and used for further characterization of the strains by dDDH and genome-genome comparison.

The text from line 183 to 187 has been deleted.

These encoded two hypothetical proteins, two IS91 insertion sequences, a disinfectant resistance gene, and four antibiotic resistance genes (dfrA19, catB3, sul1, and ant(2'')-Ia) (Fig. 1). At the phenotypic level, the absence of ant(2'')-Ia in Ciw5.2 led to the strain being sensitive towards gentamicin, while Ciw5.1 showed resistance to this antibiotic. The chloramphenicol MIC value for both strains was very high (>256 μ g/ml), because of catA1 gene, which was present in both strains (Table 2).

The text from line 203 to 210 has been deleted.

Except for strains Ciw5.1 and Ciw5.2, all Citrobacter plasmids were rather large plasmids (> 32,000 bp). Cipo13 was the only strain harboring five plasmids, which were all considered large, amounting to a total of 0.6 Mbp of extrachromosomal DNA size (Table 3). All of the Enterobacter and Escherichia strains, except for Ec1115, also possessed predominantly large plasmids. Strain Ec1115 had one small as well as three large plasmids. The K. pneumoniae strains had a variety of plasmid sizes, including generally one large and one small plasmid (Table 3).

6. In the discussion section, please provide possible explanations for your major findings while comparing these to the findings of other relevant studies.

Line 365

This finding in the context of gene transferring suggested that genetic transferring events are expected to occur in both directions between chromosomes and plasmid.

Line 407

Furthermore, these complete plasmid sequences were utilized not only to classify the replication types of plasmids but also to analyze the entire set of gene features associated with plasmid functions, in comparison to the previous study (19).

7. Please highlight some limitations of the present study in the last paragraph of the discussion section.

Thank you for this suggestion, the following paragraph was added:

In this study, we focused on 12 species belonged to *Enterobacteriaceae* from fresh produce in northern Germany and generated only 22 complete genome sequences. However, the microbiota of fresh produce, including agricultural soil and irrigation water, are much more complex and also highly diverse. Furthermore, while we conducted complete genome sequences, we did not enumerate antibiotic-resistant bacteria present on the fresh produce. Therefore, the 12 species considered in this study may not provide sufficient data on the potential of antibiotic resistance plasmid spread from fresh produce and more extensive research on this would clearly be required.

8. Please provide some recommendations based on the key findings of this study in the last paragraph of the conclusion.

Please see above.

Re: Spectrum00361-24R1 (Antibiotic-resistance plasmids in Enterobacteriaceae isolated from fresh produce in northern Germany)

Dear Dr. Gyu-Sung Cho:

Thank you for providing a point-by-point response to the concerns of the reviewers. However, like reviewer 2 mentioned, the manuscript is still unnecessarily too long and should be markedly reduced. I have made suggestions on areas of the manuscript that can be removed.

Revision Guidelines

Sincerely,
Adelumola Oladeinde
Editor
Microbiology Spectrum

Lines 106 - 110: Since this is the first mention of the strains can the authors use the complete genus name. For instance, what is En? It could be Enterococcus or Enterobacter.

Lines 114 - 118 : Please delete as these results were also reported under "genomic characterization" where it is a better fit.

Lines 280 - 287: There is no context provided to support the results of Plasmid Copy Number. I will strongly suggest that the

authors remove this part of the results. It does not add any value to the manuscript and seems out of place.

Lines 301 - 322: I will suggest the authors focus the discussion on results that are associated with the objective of their study i.e., AMR. Furthermore, this observation of diversity in genome sizes was mentioned in the M&M/results which is sufficient and makes the discussion redundant. There is no need to dedicate a page to discuss this. To reduce the length of this manuscript, I will recommend that the authors remove lines 301 -322.

Lines 344 - 346: This has been mentioned in the M&M and results. Please remove because it is redundant.

Lines 358 - 362: Not clear "what was not the case" in your study. That you did not find IncF or that you found IncI1. Please re-write for clarity.

Lines 366 - 367: Not sure what the authors mean by " could determine the presence e.g., of IncF, IncU and IncN replicons. Please can this be written in a complete sentence.

Line 380: Add "e" after pneumonia (e)

Lines 380 - 381: While this is possible (blaCTX and antibiotic treatment), a lot of biological and physiological processes will have to occur at this time. Therefore, the authors are just guessing. I will suggest the authors remove this sentence.

Lines 399 - 405: Like I mentioned earlier, please use the Plasmid Copy Number results and discussion for another study. Please remove.

Line 415: add that before "belonged"

Line 416: remove "only"

Line 421: Remove "clearly"

Table 1: Foot note; Freundii is missing a "n".

Table 5 : Please remove

Lines 528 - 558 : Please delete

Response to reviewers

Lines 106 - 110: Since this is the first mention of the strains can the authors use the complete genus name. For instance, what is En? It could be Enterococcus or Enterobacter.

Done as suggested. The sentence was as follows.

The sequencing by both long and short read methods suggested that strains Enterobacter (En.) dykesii Endy1 and En. dykesii Endy2 were very similar, and dDDH values showed 100 % identity. In addition, both strains were also isolated from the same fresh produce (Table 1) and therefore these appeared to probably represent clonal isolates. Strain Citrobacter (C.) gillenii Cigi1 was compared by dDDH.....

Lines 114 - 118 : Please delete as these results were also reported under "genomic characterization" where it is a better fit.

Done as suggested.

Deleted this paragraph. "Despite showing 100 % identity when compared using dDDH and having identical plasmids, C. werkmanii Ciw5.1 showed an approximately 8809-bp larger chromosome than C. werkmanii Ciw5.2. Notably, a large antibiotic resistant gene region was present in strain Ciw5.1 but not in Ciw5.2, and this region and surrounding sequences are indicated using Easyfig (v.2.2.2) (Fig. 1). "

Lines 280 - 287: There is no context provided to support the results of Plasmid Copy Number. I will strongly suggest that the authors remove this part of the results. It does not add any value to the manuscript and seems out of place.

Done as suggested.

Deleted this paragraph. "The copy numbers of specific, potentially conjugative plasmids carrying antibiotic-resistance genes present in each of the strains Kpneu28, Kpneu34, Ec1115 and Ec1117 was determined using qPCR. The copy numbers of the four IncF plasmids were between 1.8 and 3.8 copies per chromosome (Table 5). The two plasmids of the Escherichia strains exhibited lower copy numbers when compared to the Klebsiella plasmids (Table 5). The smallest tested plasmid (pEC1117_1) (119,797 bp) had the lowest determined copy number of 1.8 / chromosome, while the plasmid with the highest copy number (3.8 copies / chromosome; pKPNEU28_2) was only slightly larger (121,449 bp) than pEC1117_1."

Lines 301 - 322: I will suggest the authors focus the discussion on results that are associated with the objective of their study i.e., AMR. Furthermore, this observation of diversity in genome sizes was mentioned in the M&M/results which is sufficient and makes the discussion redundant. There is no need to dedicate a page to discuss this. To reduce the length of this manuscript, I will recommend that the authors remove lines 301 -322.

Done as suggested.

Deleted this paragraph.

“The strain pairs Cipa6.1 and Cipa6.2, Ciw5.1 and Ciw5.2 and Endy1 and Endy2 were very similar to each other and the dDDH values indicated 100% similarity. Therefore, these strain pairs respectively appeared to represent clonal isolates. Despite also showing 100 % identity when compared using dDDH and having identical plasmids, Ciw5.1 showed an approximately 8809 bp larger chromosome when compared to Ciw5.2. This was found to be due to a region with an integron integrase and other genetic elements that included transposases, hypothetical proteins, two IS91 and antibiotic resistance genes. This region of the chromosome shows multiple transposases and at least one integron integrase, which makes it a perfect absorber of the genes that were acquired. Integrons are an ideal spot for adding genes, thereby increasing the diversity to the genome while minimally disturbing the chromosome [52]. Since both strains were isolated from the same product sample, it is hypothesized that the strains have a common ancestor and hypothetically either Ciw5.1 gained the extra DNA sequence, or Ciw5.2 lost the genes located on the 8152-bp DNA fragment as an adaptive microevolutionary change. Either way, it exemplifies the high potential for genetic exchange that exists in the bacteria present on fresh produce. This finding in the context of gene transferring suggested that genetic transferring events are expected to occur in both directions between chromosomes and plasmids. “

Lines 344 - 346: This has been mentioned in the M&M and results. Please remove because it is redundant.

Done as suggested.

Deleted this paragraph.

“There was no apparent relationship between size and mol% GC content of the plasmids and plasmids ranged from 2,058 bp to 326,286 bp. Other studies have also reported on high variability on plasmid sizes of Enterobacteriaceae (37-39), indicating plasmid sizes from 1,100 bp to 404,000 bp (38).”

Lines 358 - 362: Not clear "what was not the case" in your study. That you did not find IncF or that you found IncI1. Please re-write for clarity.

Thank you for pointing this out.

Now the sentence is changed as follows.

In this study, only one plasmid sequence of E.coli (pEC1115_2) had the IncI1 replicon type, in contrast to the study by Blau et al (19).

Lines 366 - 367: Not sure what the authors mean by " could determine the presence e.g., of IncF, IncU and IncN replicons. Please can this be written in a complete sentence.

Now the sentence is changed as follows.

Huizinga et al. (37) investigated extended spectrum β -lactamase producing bacteria from sprouts in the Netherlands and also found Col and IncF replicon types to predominate in whole genome sequence data using PlasmidFinder (v. 2.1). They also identified the presence of replicons such as IncF, IncU and IncN.

Line 380: Add "e" after pneumonia (e)

It is changed.

Lines 380 - 381: While this is possible (blaCTX and antibiotic treatment), a lot of biological and physiological processes will have to occur at this time. Therefore, the authors are just guessing. I will suggest the authors remove this sentence.

It is deleted as suggested.

Lines 399 - 405: Like I mentioned earlier, please use the Plasmid Copy Number results and discussion for another study. Please remove.

It is deleted as suggested.

Line 415: add that before "belonged"

It is added as suggested.

Line 416: remove "only"

It is removed.

Line 421: Remove "clearly"

It is removed.

Table 1: Foot note; Freundii is missing a "n".

it is added.

Table 5 : Please remove

It's done.

Lines 528 - 558 : Please delete

It is deleted.

Re: Spectrum00361-24R2 (Antibiotic-resistance plasmids in Enterobacteriaceae isolated from fresh produce in northern Germany)

Dear Dr. Gyu-Sung Cho:

Thank you for your patience throughout the review. Your manuscript has been accepted and I am forwarding it to the ASM production staff for publication.

I will recommend that the authors add more subsections to the results under genome characteristic before the ASM production staff begin working on the proof. As it is, this subsection is five pages long, making it difficult for the reader to follow.

Based on my reading, I found three possible sub-sections:

Genome characteristics: Lines 129 - 181

AMR: Lines 182 - 226

Plasmids/Transposons/Integrans: Lines 227 - 276

Your paper will first be checked to make sure all elements meet the technical requirements. ASM staff will contact you if anything needs to be revised before copyediting and production can begin. Otherwise, you will be notified when your proofs are ready to be viewed.

Sincerely,
Adelumola Oladeinde
Editor
Microbiology Spectrum